# Designing reliable and accurate isotope-tracer experiments for CO$_2$ photoreduction

Shengyao Wang [1,2,3], Bo Jiang [4] ✉, Joel Henzie [2], Feiyan Xu [5], Chengyuan Liu [6], Xianguang Meng [2,7], Sirong Zou [1], Hui Song [2], Yang Pan [6], Hexing Li [4], Jiaguo Yu [5] ✉, Hao Chen [1] ✉ & Jinhua Ye [2,8,9] ✉

The photoreduction of carbon dioxide (CO$_2$) into renewable synthetic fuels is an attractive approach for generating alternative energy feedstocks that may compete with and eventually displace fossil fuels. However, it is challenging to accurately trace the products of CO$_2$ photoreduction on account of the poor conversion efficiency of these reactions and the imperceptible introduced carbon contamination. Isotope-tracing experiments have been used to solve this problem, but they frequently yield false-positive results because of improper experimental execution and, in some cases, insufficient rigor. Thus, it is imperative that accurate and effective strategies for evaluating various potential products of CO$_2$ photoreduction are developed for the field. Herein, we experimentally demonstrate that the contemporary approach toward isotope-tracing experiments in CO$_2$ photoreduction is not necessarily rigorous. Several examples of where pitfalls and misunderstandings arise, consequently making isotope product traceability difficult, are demonstrated. Further, we develop and describe standard guidelines for isotope-tracing experiments in CO$_2$ photoreduction reactions and then verify the procedure using some reported photoreduction systems.

If sunlight can be efficiently harnessed as a primary energy source for the photoreduction of CO$_2$ into synthetic fuels, it will contribute a large flow of carbon-neutral energy in a material format that is compatible with existing fossil fuel infrastructure[1–8]. At a large enough scale, this technology may enable the control of atmospheric CO$_2$ levels and make fossil fuels obsolete in the coming decades[9–12]. However, considerable scientific and technical challenges must be overcome to realize CO$_2$ photoreduction reactions with high selectivity and conversion efficiency[13–18]. After many decades of exploration, most experimental studies focus on optimizing materials and reaction systems[19–24]. Most reported CO$_2$ photoreduction systems are not yet able to deliver the desired products on a practical scale and with long-term operational stability. The low selectivity and CO$_2$ conversion efficiency of most existing reaction systems[25] is a significant roadblock

[1]College of Science, Shenzhen Institute of Nutrition and Health, Huazhong Agricultural University, 430070 Wuhan, P. R. China. [2]International Center for Materials Nanoarchitectonics (WPI-MANA), National Institute for Materials Science (NIMS), 1-1 Namiki, Tsukuba, Ibaraki 305-0044, Japan. [3]Shenzhen Branch, Guangdong Laboratory for Lingnan Modern Agriculture, Genome Analysis Laboratory of the Ministry of Agriculture, Agricultural Genomics Institute at Shenzhen, Chinese Academy of Agricultural Sciences, 518120 Shenzhen, P. R. China. [4]The Education Ministry Key Lab of Resource Chemistry, Joint International Research Laboratory of Resource Chemistry, Shanghai Frontiers Science Center of Biomimetic Catalysis, College of Chemistry and Materials Science, Shanghai Normal University, 200234 Shanghai, China. [5]Laboratory of Solar Fuel, Faculty of Materials Science and Chemistry, China University of Geosciences, 430074 Wuhan, P. R. China. [6]National Synchrotron Radiation Laboratory, University of Science and Technology of China, 230029 Hefei, P. R. China. [7]Hebei Provincial Key Laboratory of Inorganic Nonmetallic Materials, College of Materials Science and Engineering, North China University of Science and Technology, 063210 Tangshan, P. R. China. [8]Graduates School of Chemical Science and Engineering, Hokkaido University, Sapporo 060-0814, Japan. [9]TU-NIMS International Collaboration Laboratory, Tianjin University, 300072 Tianjin, P. R. China. ✉e-mail: jiangbo@shnu.edu.cn; yujiaguo93@cug.edu.cn; hchenhao@mail.hzau.edu.cn; Jinhua.YE@nims.go.jp

in this field because it is challenging to attribute potential products (e.g., CO, alkanes, alcohols, carboxylic acids, and alkenes) solely to $CO_2$ photoreduction processes[26,27]. Unfortunately, the efficiency of some high-performance reaction systems is later attributed to the decomposition of the carbon contaminants on the photocatalyst or the reaction system[28–32].

Increasing awareness of the challenge of attributing products to $CO_2$ photoreduction has made $^{13}C$ isotope labeling experiments essential[33–35]. Lehn et al. carried out isotope labeling experiments with $^{13}CO_2$ in 1983, revealing that the reduction product of CO indeed came from $CO_2$ by using gas chromatography-mass spectrometry (GC-MS). However, no standard test method was mentioned in this research[36,37]. Later, Willner et al. used $^{13}C$ nuclear magnetic resonance spectroscopy (NMR) to show that $H^{13}COO^-$ originates from $H^{13}CO_3^-$ (the dissolved $^{13}CO_2$ in solution)[38]. Over the years, various techniques using NMR spectroscopy and Fourier transform infrared spectroscopy (FT-IR) have been employed in isotope-tracer studies: the approach of $^1H$-NMR is efficient for liquid product analysis through peak shifts and coupling constants of the $^{13}C$-linked hydrogen; FT-IR equipped with a gas cell is effective for gaseous product analysis via the increased path length of a beam by multiple internal reflections[39,40]. Still, both methods are relatively insensitive to various carbon isotopes. Thus GC-MS-based techniques are still the most reliable and universal strategy for isotope-tracer studies in $CO_2$ photoreduction[41,42]. For example, numerous organic carbon-containing photocatalysts, such as conjugated polymers, Metal-organic frameworks (MOFs), and Covalent organic framework (COFs), have good activities in $CO_2$ photoreduction[43–48]. Isotope-tracing studies with GC-MS allow researchers to exclude carbon contamination from the decomposition of these catalysts or the materials synthesis process. Current isotope-tracer methods can measure simple samples such as $^{13}CO$, $H^{13}COOH$, $^{13}CH_4$, $^{13}C_2H_6$, $^{13}C_2H_4$, and $^{13}CH_3OH$ due to the isotope effects induced by different mass-to-charge ratios[49–52]. However, once the abovementioned series of molecular substances are generated during $CO_2$ photoreduction, the inherent ionization process of mass detection could not only charge the molecules but also inevitably cause chemical bonds to break in these molecules, making the isotope-tracing via mass detection becomes increasingly difficult as molecular fragments with similar $m/z$ ratios interfere and lead to misidentification and errors in quantification. Even for the emerging synchrotron vacuum ultraviolet photoionization mass spectrometry (SVUV-PIMS) strategy, the existence of interfering factors induced by the long-time sampling is also uncertain[53]. So the gas chromatograph (GC) is essential to be added to the sampling system to separate molecular species before collecting mass spectra (MS)[54,55].

Despite the fact that isotope-tracer experiments are regarded as solid corroborating evidence for $CO_2$ photoreduction, the current protocols are still substandard, causing the literature to be rife with false-positive results[56]. Meanwhile, the accurate and effective solution of isotope detection for various products in $CO_2$ photoreduction is still lacking, which causes the community to be suspicious of all results. In the present work, we confirmed the pros and cons of the current mass spectrometry protocol as well as the emerging technology of synchrotron vacuum ultraviolet photoionization mass spectrometry (SVUV-PIMS) when they applied to the isotope-tracer experiments in $CO_2$ photoreduction. We also experimentally performed isotope-tracer experiments on various isotope standards and described a standard protocol assessing various potential products of $CO_2$ reduction. In addition, some classic $CO_2$ photoreduction systems reported in the literature were also used to validate our protocols. It is imperative to settle on an appropriate scientific method for isotope-tracer studies to promote trust in the community as we develop and benchmark some materials systems for $CO_2$ photoreduction. This research illustrates the pitfalls and misunderstandings in isotope-tracer studies and also provides examples and references for $^{13}C$ isotope-tracer methods so we can all obtain solid evidence that reduction products are firmly attributed to $CO_2$ photoreduction reactions.

## Results

### Importance of separation for multicomponent samples

GC-MS is the most common instrument for tracing isotopes in $CO_2$ reduction reactions currently. The GC-MS instrument is composed of a GC to separate molecular species based on affinity with a column material and an MS to detect their mass or their mass fragments (Fig. 1a). However, the current method for tracing isotopes in $CO_2$ reduction reactions ignores this. We verified the reliability of the current method. To mimic the products and conditions of a real photoreduction reaction, we directly employed standard gases such as $^{13}CO_2$, $^{13}CO$, and vapor in the $CO_2$ reduction system. The volume ratio of $^{13}CO$ >5% v/v is much higher than the yield of CO in the usual $CO_2$ photoreduction process. Then these premixed gases were collected from the $CO_2$ reduction system and injected into the GC-MS equipped with a commercial HP-5ms column (see "Methods" for experimental details). As shown in Fig. 1b, a peak at $m/z = 29$ in the mass spectrogram could be obtained from TIC at an RT of 6.4 min. This peak has a definite value, but it could originate from three sources: (i) it is a molecular ion of $^{13}CO$ ($^{13}CO^+$, $m/z = 29$), (ii) a fragment ion of $^{13}CO_2$ ($^{13}CO^+$, $m/z = 29$), or (iii) it is a molecular ion of $^{15}N^{14}N$ (the natural abundance of nitrogen isotope, $^{15}N^{14}N^+$, $m/z = 29$). This could be further confirmed via the GC-MS analysis of pure $^{13}CO_2$, $^{13}CO$, and $N_2$ (Supplementary Figs. 1–3). Although the natural abundance of the nitrogen isotope exists in traces, the $^{13}CO_2$ fragment ion is abundant and has the same characteristics to interfere with the isotope-tracer results for $^{13}CO$. Consequently, even if there is no $^{13}CO$ in the injected mixture (only $^{13}CO_2$ and vapor were injected into GC-MS), a similar mass spectrogram can be obtained under the same conditions (Fig. 1c), generating a peak at $m/z = 29$. This peak could be mainly attributed to the fragment ion ($^{13}CO^+$) generated from the dissociation of $^{13}CO_2$. Even the deactivated fused silica tube (the length is 5 m), without any separation effect, acts as the connector to let all components ($^{13}CO_2$ and vapor) enter the quadrupole together; a similar result can also be obtained in the mixture of $^{13}CO_2$ and vapor even without a photocatalytic reduction process (Supplementary Fig. 4). Moreover, the selected $m/z = 45$, 29 and 17 can be detected (Supplementary Fig. 5) when the detection method of GC-MS is set to the selected ion monitoring (SIM) mode (see "Methods" for experimental details). It is always used as evidence of traceability for the products originating from $CO_2$ photoreduction, and signals at $m/z = 17$, 29, and 45 are assigned to $CH_4$, CO, and $CO_2$ molecular ions, respectively[57]. However, based on the above analysis, the presence of dissociated $^{13}CO_2$ (generated fragments of $^{13}CO^+$, $m/z = 29$) and vapor (generated fragments of $HO^+$, $m/z = 17$) throws these conclusions of traceability into doubt.

For such confusing results, we could ascribe them to the detection principle of MS detector for charged species. Namely, the process of charging molecules in an MS detector is not so controllable, and there is some probability that the dissociation of the molecules coincides with ionization. In particular, when the amount of the sample is large, the probabilistic dissociation and dissociative recombination become inevitable, causing molecular ions and fragmented ions with the same characteristics to interfere with each other on the MS detector[57]. As a result, confusion could occur in the isotope detection for the product of $^{13}CO$ due to the simultaneous process of ionization and dissociation over the $^{13}CO_2$ and $H_2O$ reactants, $^{13}CO$ product, and air components ($N_2$, $O_2$, Ar), respectively (Fig. 2a).

To avoid this confusion, suitable chromatographic conditions are a prerequisite to meeting the requirement of sample separation. As shown in Fig. 2b, a GC separator equipped with a suitable chromatographic column can separate mixed gases of $CO_2$ and CO so that they

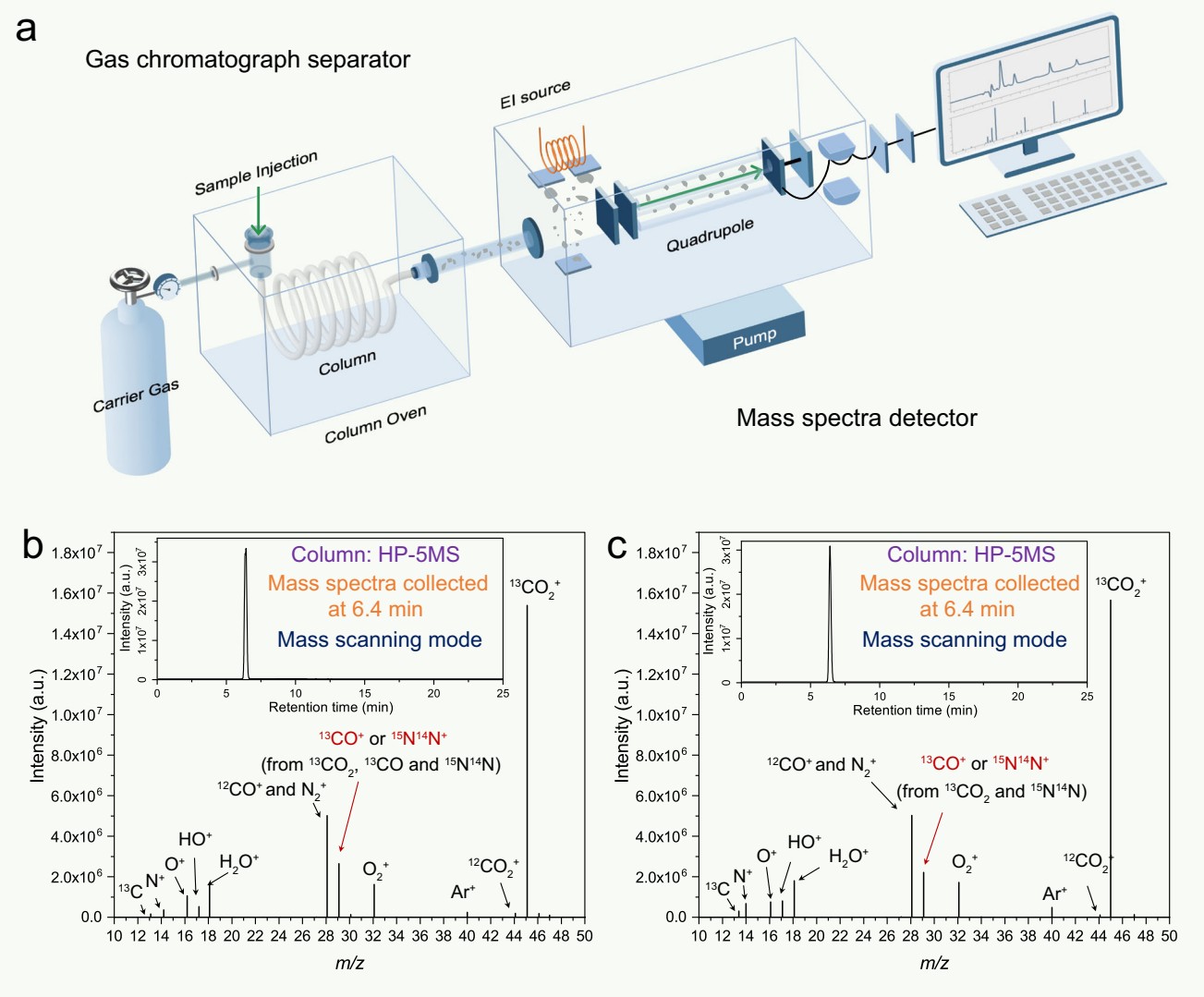

**Fig. 1 | The contemporary approach toward isotope-tracing experiments in $CO_2$ photoreduction. a** An illustration of a GC-MS with the GC separator and MS detector used to trace isotopes in $CO_2$ reduction. **b**, **c** MS spectra and TIC (inset) using an HP-5ms column for the model gases collected from the sealed $CO_2$ reduction system: **b** after photoreduction ($^{13}CO_2$, $^{13}CO$, and vapor) and **c** before photoreduction ($^{13}CO_2$ and vapor). Source data are provided as a Source Data file.

are ideally queued in a single file before entering the MS detector. Thus although both $CO_2$ and CO can generate the same fragment ions in the ion source, their different entry times provide resolution. As a result, two signal peaks appeared at different retention times (RT) in the total ion chromatography (TIC). Their corresponding mass spectrogram ($MS_1$ and $MS_2$ selected at different RT) show both molecular ions and fragment ions, which can be clearly attributed to either CO or $CO_2$. In contrast, chromatography columns with bad separation characteristics allow components of $CO_2$ and CO to enter the ion source simultaneously and generate complex molecular ions and fragment ions. These then enter the quadrupole together, resulting in a single peak on TIC and a corresponding mass spectrogram (MS). This spectrogram is very close to the abovementioned $MS_1$ obtained from the fully separated samples. However, the signal corresponding to $CO^+$ could be attributed to a fragment ion of $CO_2$ or a molecular ion of CO, making it impossible to distinguish the source of $CO^+$. Similarly, it is impossible to attribute the signal from $C^+$ or $O^+$ because both $CO_2$ and CO can generate it.

Apart from the abovementioned method of mass spectrometry for isotope traceability, recently, an emerging technique called the synchrotron vacuum ultraviolet photoionization mass spectrometry (SVUV-PIMS) was also used to assign CO products to $CO_2$ photoreduction reactions[58]. This analytical technique is useful because it delivers high-intensity, high-resolution, and widely tunable photon energies for photoionization processes. SVUV-PIMS enables selective and sensitive ionization (see "Methods" for experimental details) and can overcome the drawbacks of other ionization techniques and minimize fragmentation interference in MS detection. As shown in Fig. 3a, the $m/z = 45$ signal gives an ionization threshold of 13.75 eV in the photoionization spectra of pure $^{13}CO_2$, while pure $^{13}CO$ exhibits an increased ionization threshold of 14.00 eV in the $m/z = 29$ signal and the response rate of CO 10 times less than that of $CO_2$. However, the SVUV-PIMS signal at $m/z = 29$ is also observed at 13.91 eV for pure $^{13}CO_2$ (Fig. 3b). This feature matches the weak fragmentation of $^{13}CO_2$ or via impurities introduced during the production of $^{13}CO_2$ gas, interfering with the detection of $^{13}CO$ to some extent. Therefore, when the photon energy of SVUV-PIMS was set at 14.50 eV to monitor the isotope product of $^{13}CO$, the peak at $m/z = 29$ can be found in the SVUV-PIMS spectra of pure $^{13}CO_2$ (Fig. 3c) that is similar to the spectra obtained from the mixture of $^{13}CO$ and $^{13}CO_2$ (inset of Fig. 3c). Moreover, the intensity of $m/z = 45$ signal in pure $^{13}CO_2$ is strongly associated with the increased repulsion voltage and sampling time (Fig. 3d). Although SVUV-PIMS minimizes fragmentation of the potential analytes, the popularity of this technique and the excessive demand of sample

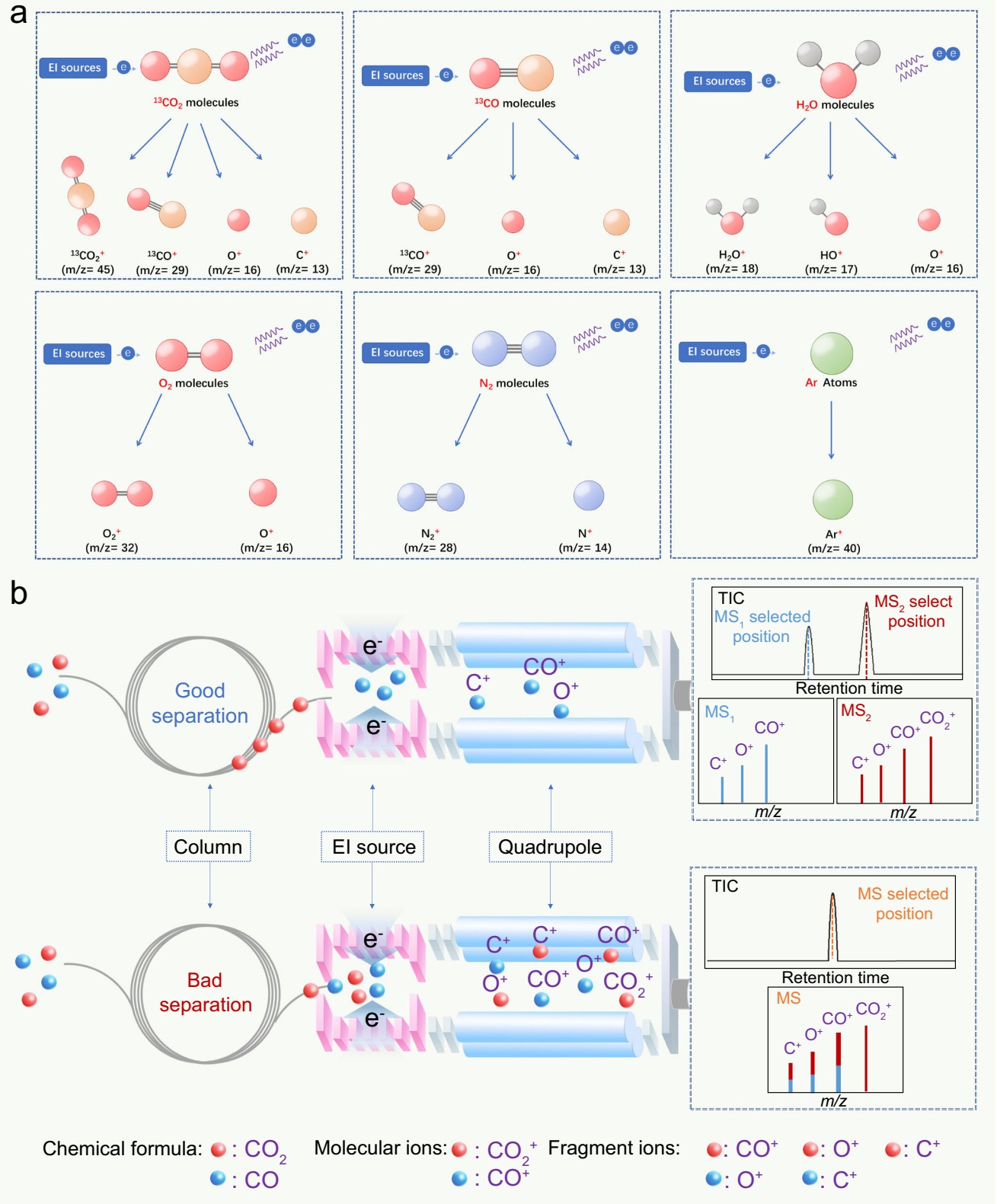

**Fig. 2 | The pitfalls and misunderstandings arise in the contemporary approach. a** An illustration of the dissociation processes as the reactants ($^{13}CO_2$, $H_2O$), product ($^{13}CO$), and air impurities ($N_2$, $O_2$, Ar) as they undergo ionization in the EI source in GC-MS. Analytical data of $^{13}C$ isotope standard samples. **b** The illustrations of good separation condition (top) and bad separation condition (bottom) in GC for MS detection when the mixture of $CO_2$ and CO to be analyzed potentially interfered with each other. Source data are provided as a Source Data file.

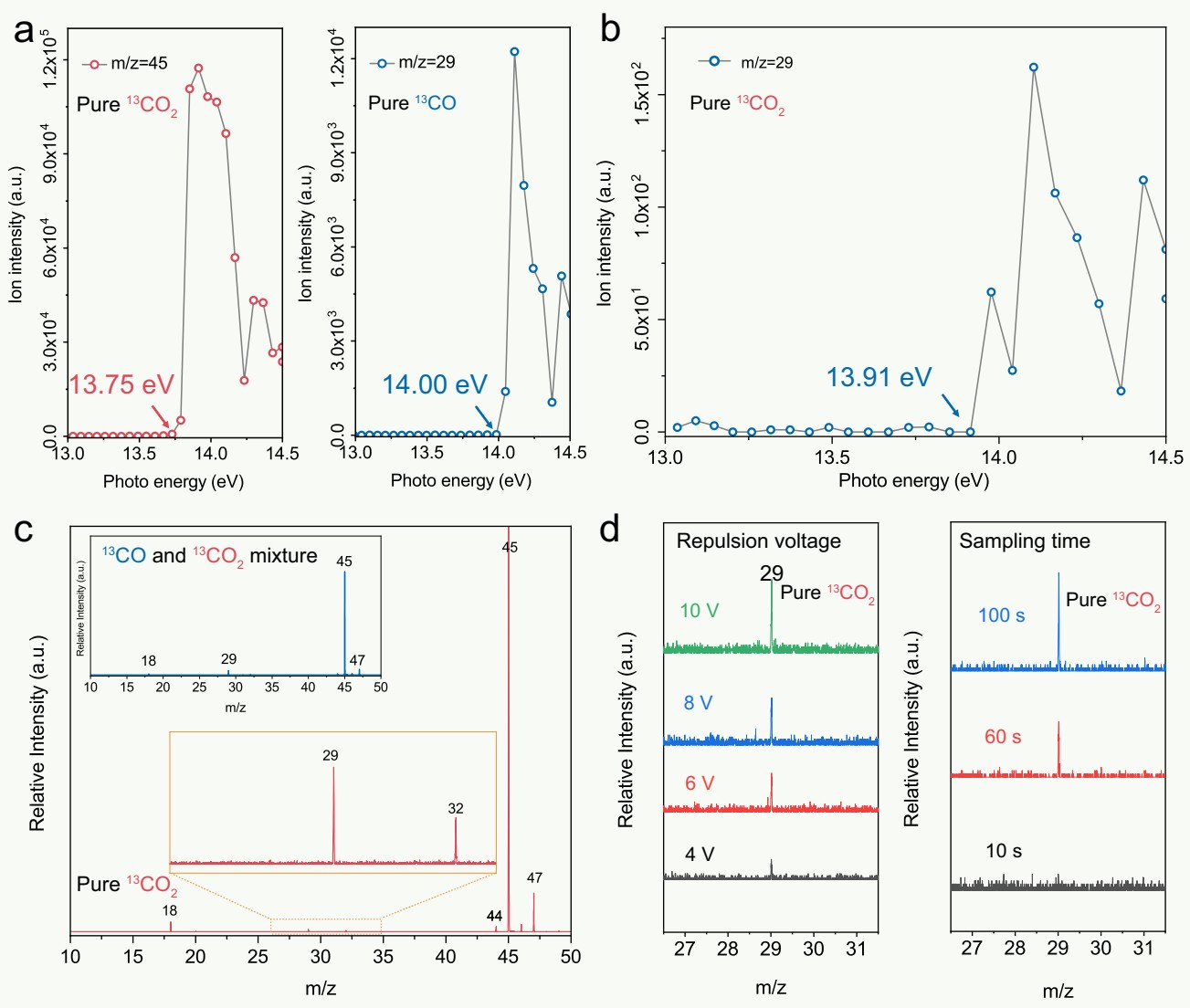

**Fig. 3 | SVUV-PIMS for isotope-tracing experiments in CO₂ photoreduction.** Photoionization efficiency spectra for **a** $m/z = 45$ at a photon energy of 13.75 eV for pure $^{13}CO_2$ and $m/z = 29$ at a photon energy of 14.00 eV for pure $^{13}CO$; and **b** $m/z = 29$ at a photon energy of 13.91 eV for pure $^{13}CO_2$. SVUV-PIMS spectra of the gas components of **c** pure $^{13}CO_2$ and a mixture of $^{13}CO$ and $^{13}CO_2$ (inset) at a photon energy of 14.5 eV. **d** SVUV-PIMS spectra at different repulsion voltages and sample times for $m/z = 29$ over pure $^{13}CO_2$ at a photon energy of 14.5 eV. Source data are provided as a Source Data file.

(hundred milliliters) hamper the extensive application of SVUV-PIMS to some extent. Moreover, the existence of interfering factors in this strategy induced by the long-time sampling is just like the SIM model in GC-MS, which cannot be ignored in the isotope-tracer experiment for CO₂ photoreduction. Therefore, a complete separation of multiple components in GC is essential to obtain reliable isotope-tracing results for the products of CO₂ photoreduction in MS.

## Analysis strategy for the isotope-labeled products of CO₂ photoreduction

Potential products of CO₂ photoreduction reactions include CO, CH₄, C₂H₆, C₂H₄, HCOOH, CH₃COOH, CH₃OH, and CH₃CH₂OH. Isotope-tracing experiments are complicated because products are inevitably mixed with the reactants (H₂O and CO₂) and impurities introduced during sampling (e.g., N₂ and O₂ via the airtight syringe). In addition, chromatographic columns such as HP-5ms in GC-MS cannot separate these gas components in isotope-tracing experiments. Thus, selecting a suitable column for separating permanent gas components is critical. For example, the HP-Molesieve column (see "Methods" for experimental details), as shown in Fig. 4a, can completely separate O₂ (RT at

3.65 min), N₂ (RT at 4.37 min), CH₄ (RT at 5.25 min), and CO (RT at 7.25 min). After complete separation, it prevents the introduced air components during the injection process from interfering with the detection of products. The obtained mass spectra of O₂ and N₂ closely match the NIST mass spectral library (Supplementary Figs. 6 and 7), and the $m/z$ of carbon-related molecular ions and fragment ions generated from $^{13}CH_4$ ($^{13}CH_4^+$, $m/z = 17$; $^{13}CH_3^+$, $m/z = 16$; $^{13}CH_2^+$, $m/z = 15$; $^{13}CH^+$, $m/z = 14$; and $^{13}C^+$, $m/z = 13$) and $^{13}CO$ ($^{13}CO$, $m/z = 29$ and $^{13}C$, $m/z = 13$) have a mass shift effect (M+1) compared with the non-isotope-labeled CH₄ (Supplementary Fig. 8) and CO (Supplementary Fig. 9), respectively. Nevertheless, the molecular sieve column will irreversibly adsorb the reactant of CO₂; thus, MS analysis of the reactant (CO₂) and the products (CO or CH₄) cannot be obtained simultaneously with the HP-Molesieve. To confirm the isotopic abundance of CO₂, the HP-PLOT/Q is employed (see "Methods" for experimental details). As shown in Fig. 4b, the CH₄ (RT at 4.87 min) and CO₂ (RT at 6.60 min) elute as separate peaks in the TIC, and the corresponding MS of $^{13}CH_4$ and $^{13}CO_2$ ($^{13}CO_2^+$, $m/z = 45$; $^{13}CO^+$, $m/z = 29$; and $^{13}C^+$, $m/z = 13$) could be obtained, respectively. The peaks exhibit a mass shift effect (M+1) compared to the non-isotope-labeled standards of CH₄

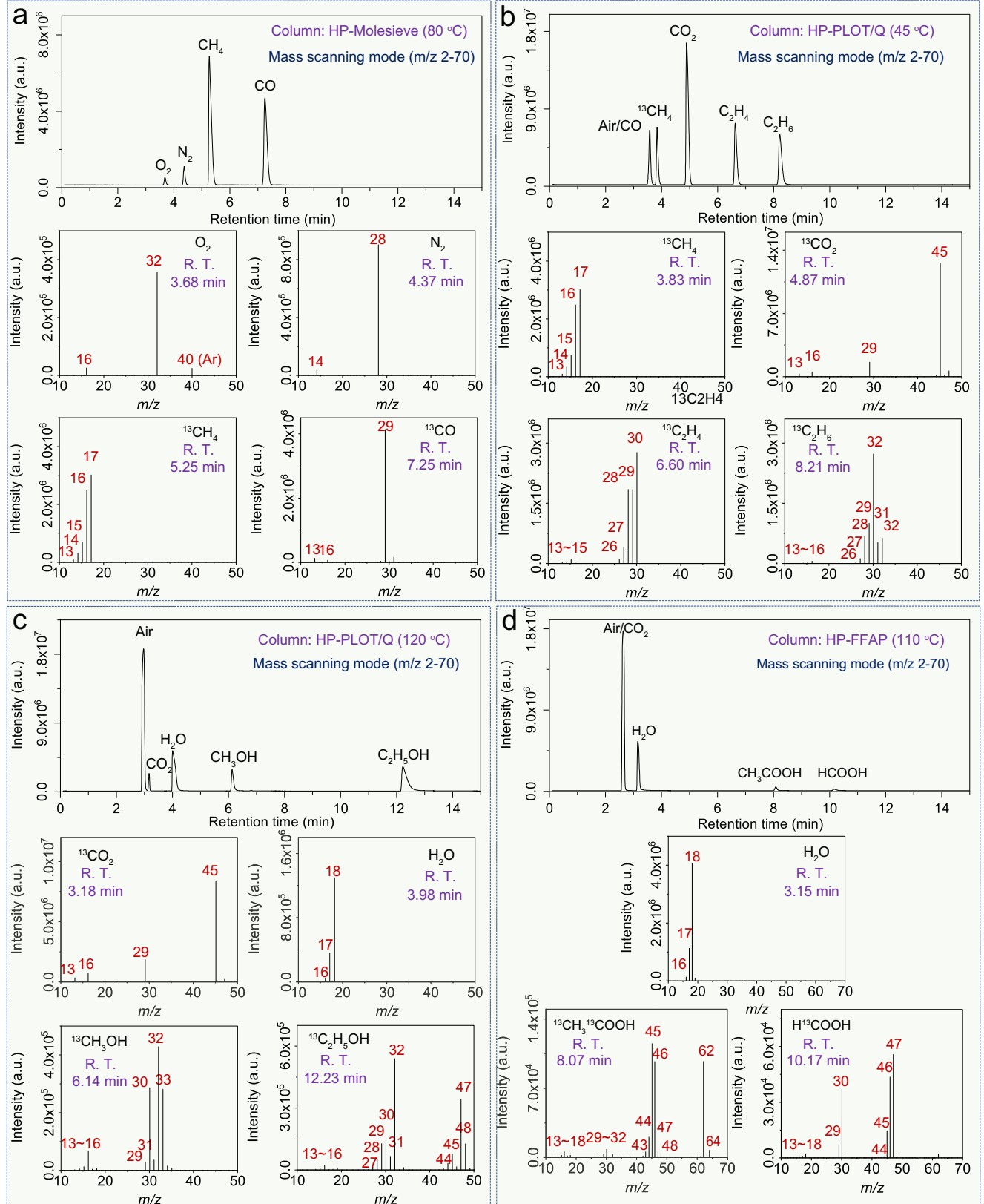

**Fig. 4 | Standard guidelines for isotope-tracing experiments in CO₂ photo-reduction.** Analytical data of ¹³C isotope standard samples. TIC and corresponding MS spectra for **a** O₂, N₂, ¹³CH₄, and ¹³CO using the HP-Molesieve column at 80 °C; **b** ¹³CH₄, ¹³CO₂, ¹³C₂H₄, and ¹³C₂H₆ using HP-PLOT/Q at 45 °C; **c** ¹³CO₂, ¹³CH₃OH, and ¹³C₂H₅OH using HP-PLOT/Q at 120 °C; and **d** ¹³CO₂, ¹³CH₃OH, and ¹³C₂H₅OH using HP-FFAP at 110 °C. Source data are provided as a Source Data file.

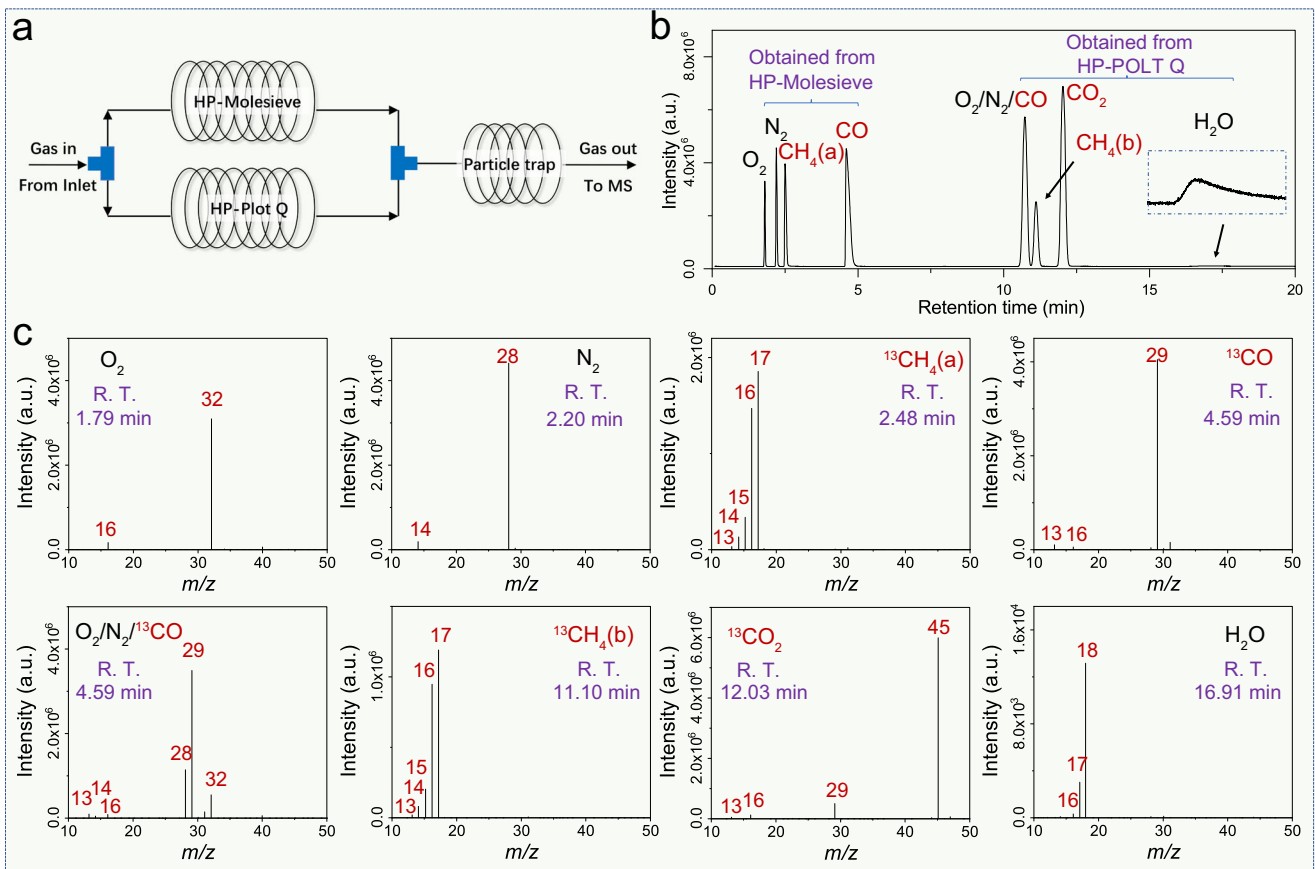

**Fig. 5 | A developed strategy of parallel connection system. a** Scheme illustrating a parallel connection system using HP-Molesieve and HP-PLOT/Q columns. The TIC (**b**) and the corresponding MS spectra (**c**) of a sample containing $O_2$, $N_2$, CO, $CH_4$, $CO_2$, and water vapor. Source data are provided as a Source Data file.

(Supplementary Fig. 8) and $CO_2$ (Supplementary Fig. 10). However, the air-derived components (e.g., $O_2$ and $N_2$) and CO (RT at 3.83 min) cannot be separated well under these conditions and may interfere with the analysis of CO isotopes. Interestingly, two kinds of $C_2$ hydrocarbons ($C_2H_6$, RT at 8.21 min and $C_2H_4$, RT at 6.60 min) could be distinguished under these conditions. Although both $^{13}C_2H_4$ and $^{13}C_2H_6$ exhibit the same highest peak at $m/z = 30$, $^{13}C_2H_6$ possesses two more characteristic peaks at $m/z = 31$ and $m/z = 32$ that can only be generated from the fragment ion of $^{13}C_2H_5^+$ and molecular ion of $^{13}C_2H_6^+$. The ratio between the peaks that recorded in MS of $^{13}C_2H_4$ ($^{13}C_2H_4^+$, $m/z = 30$; $^{13}C_2H_3^+$, $m/z = 29$; $^{13}C_2H_2^+$, $m/z = 28$; $^{13}C_2H^+$, $m/z = 27$; and $^{13}C_2^+$ $m/z = 26$) and $^{13}C_2H_6$ ($^{13}C_2H_6^+$, $m/z = 32$; $^{13}C_2H_5^+$, $m/z = 31$; $^{13}C_2H_4^+$, $m/z = 30$; $^{13}C_2H_3^+$, $m/z = 29$; $^{13}C_2H_2^+$, $m/z = 28$; $^{13}C_2H^+$, $m/z = 27$; and $^{13}C_2^+$, $m/z = 26$) is different. It is still consistent with the ratio of the non-isotope-labeled standards $C_2H_4$ (Supplementary Fig. 11) and $C_2H_6$ (Supplementary Fig. 12), respectively, which comply with the mass shift effect ($M$+2) caused by two labeled carbon atoms. In addition, another set of peaks that recorded from $m/z = 13$ to 16 exist both in the MS of $^{13}C_2H_4$ ($^{13}CH_2^+$, $m/z = 15$; $^{13}CH^+$, $m/z = 14$; $^{13}C^+$, $m/z = 13$) and $^{13}C_2H_6$ ($^{13}CH_3^+$, $m/z = 16$; $^{13}CH_2^+$, $m/z = 15$; $^{13}CH^+$, $m/z = 14$; $^{13}C^+$, $m/z = 13$), which can be attributed to fragments caused by broken carbon–carbon bonds.

When the potential products of $CO_2$ photoreduction are liquids such as alcohols or acids, we used the headspace injection method instead of airtight needles (see "Methods" for experimental details). The temperature of the HP-PLOT/Q column was raised to 120 °C to separate alcohols. The liquids suspected to be methanol or $CH_3OH$ (RT at 6.14 min) and ethanol or $CH_3CH_2OH$ (RT at 12.23 min) eluted after the air components (RT at 2.96 min), $CO_2$ (RT at 3.18 min), and $H_2O$ (RT at 3.98 min) in TIC (Fig. 4c). The material at RT = 6.14 min was identified as $^{13}CH_3OH$ based on two sets of peaks: one set at $m/z = 29$ to 33 and

another set at $m/z = 13$ to 16 attributed to the molecular ions of $^{13}CH_3OH^+$ ($m/z = 33$) and the fragment ions of $^{13}CH_3O^+$ ($m/z = 32$), $^{13}CH_2O^+$ ($m/z = 31$), $^{13}CHO^+$ ($m/z = 30$), $^{13}CHO^+$ ($m/z = 29$), $^{13}CH_3^+$ ($m/z = 16$), $^{13}CH_2^+$ ($m/z = 15$), $^{13}CH^+$ ($m/z = 14$), and $^{13}C^+$ ($m/z = 13$), respectively. It is important to note that the largest peak in the MS was $^{13}CH_3O^+$ ($m/z = 32$) due to the easier dissociation of $CH_3OH$. It also exhibits a mass shift effect ($M$+1) compared to the non-isotope-labeled $CH_3OH$ (Supplementary Fig. 13). Despite the fact that $^{13}CH_3^{13}CH_2OH$ also has its largest fragment ion peak at $m/z = 32$, the source of this peak is derived from the cleavage of the carbon–carbon bonds, generating hydrogenated $^{13}CH_2OH^+$ instead of $^{13}CH_3O^+$ as was the case for $^{13}CH_3OH$. Moreover, there is no distinct peak at $m/z = 33$ in the MS of $^{13}CH_3^{13}CH_2OH$ that matches $^{13}CH_3OH$. The MS of the ethanol fraction $^{13}CH_3^{13}CH_2OH$ also has a set of peaks at $m/z$ from 43 to 48, matching the molecular ions of $^{13}C_2H_5OH^+$ ($m/z = 48$) and fragment ions of $^{13}C_2H_4O^+$ ($m/z = 47$), $^{13}C_2H_3O^+$ ($m/z = 46$), $^{13}C_2H_2O^+$ ($m/z = 45$), $^{13}C_2HO^+$ ($m/z = 44$), and $^{13}C_2O^+$ ($m/z = 43$). It exhibited a mass shift effect of ($M$+2) or ($M$+1) versus the non-isotope-labeled $C_2H_5OH$ (Supplementary Fig. 14) and the single-isotope-labeled $CH_3^{13}CH_2OH$ (Supplementary Fig. 15), respectively, due to the two isotope-labeled carbon atoms in one molecule of $^{13}CH_3^{13}CH_2OH$ (Supplementary Fig. 16). All of these points mentioned above are sufficient enough to match the qualitative requirements for $^{13}CH_3OH$ and $^{13}CH_3^{13}CH_2OH$. Likewise, the potential products of carboxylic acids (HCOOH and $CH_3COOH$) could be separated by choosing a column (HP-FFAP) with stronger polarity (see "Methods" for experimental details). As shown in Fig. 4d, separation depends on the different molecular polarities between $CH_3COOH$, HCOOH, and the reactants of $H_2O$ and $CO_2$. $CH_3COOH$ has a relatively short retention time (RT at 8.07 min) than HCOOH (RT at 10.17 min), which elutes from the column after the air components (RT at

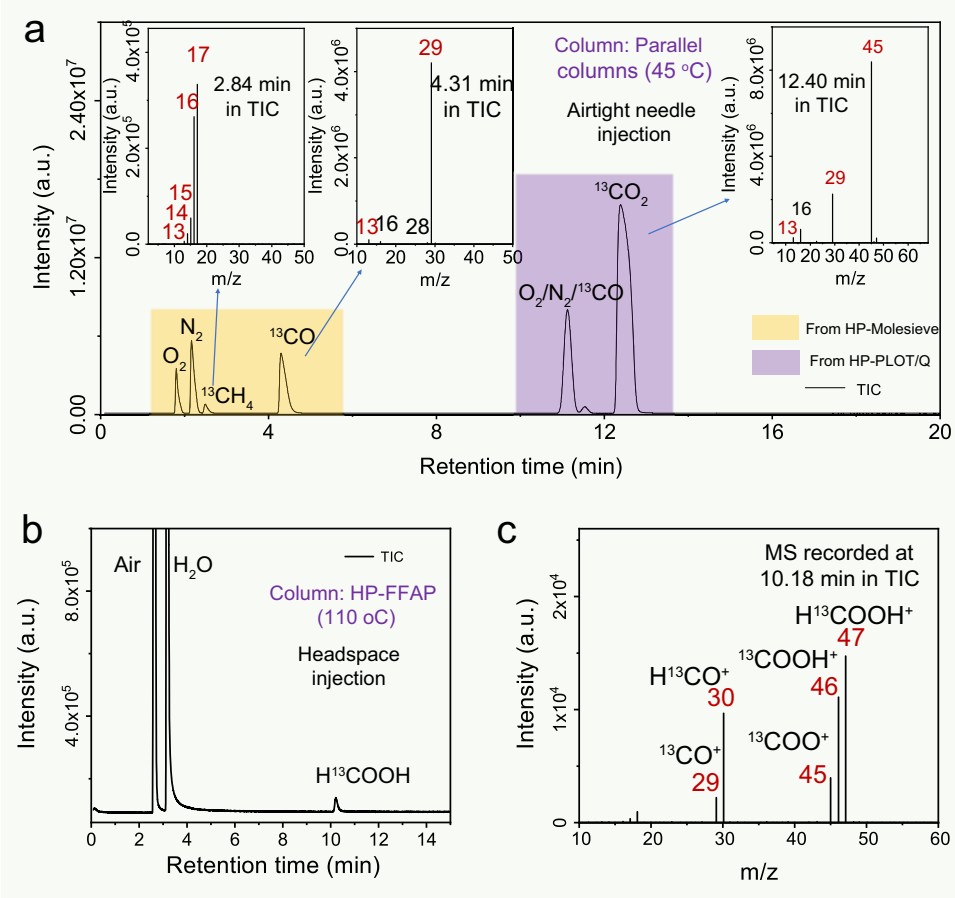

**Fig. 6 | Isotope-tracing experiments in reported homogeneous and heterogeneous photoreduction systems.** GC-MS analytical data for $^{13}C$ isotope-tracing experiments in $CO_2$ photoreduction using the following catalysts: $Fe^{III}$ porphyrin complexes in homogeneous phase (**a**) and Cd-doped ZnS in heterogeneous phase (**b**, **c**). Each panel shows the TICs and MS spectra of products generated by the catalysts in a $CO_2$ photoreduction reaction. Source data are provided as a Source Data file.

2.62 min) and $H_2O$ (RT at 3.15 min). By analyzing the corresponding MS, the non-isotope-labeled $CH_3COOH$ (Supplementary Fig. 17) possesses a molecular ion peak at $m/z = 60$ and two prominent fragment ion peaks at $m/z = 43$ and $m/z = 45$, while the corresponding molecular ion and fragment ions peak obtained from isotope-labeled $^{13}CH_3{}^{13}COOH$ shift to $m/z = 62$, $m/z = 45$, and $m/z = 46$, respectively. It reveals a difference in relative proportion compared to non-isotope-labeled $CH_3COOH$, even considering the isotope-induced mass shift effect ($M+2$). After checking the MS of $CH_3{}^{13}COOH$ (Supplementary Fig. 18), a molecular ion peak at $m/z = 61$ and two fragment ion peaks at $m/z = 44$ and $m/z = 46$ were obtained. The two fragment ions of $CH_3{}^{13}COOH$ ($m/z = 44$ and $m/z = 46$), as well as the corresponding fragment ions of $^{13}CH_3{}^{13}COOH$ ($m/z = 45$ and $m/z = 46$) and $CH_3COOH$ ($m/z = 43$ and $m/z = 45$), are generated by carbon–carbon bond cleavage ($COOH^+$) and dehydroxylation ($CH_3CO^+$), respectively (Supplementary Fig. 19). In addition, the isotope-labeled $H^{13}COOH$ possesses $H^{13}COOH^+$ ($m/z = 47$) molecular ions and fragment ions including $^{13}COOH^+$ ($m/z = 46$), $^{13}COO^+$ ($m/z = 45$), $H^{13}CO^+$ ($m/z = 30$), and $^{13}CO^+$ ($m/z = 29$), which exhibit a mass shift effect ($M+1$) compared to the non-isotope-labeled HCOOH (Supplementary Fig. 20).

The four methods mentioned above can identify reactants and products in $CO_2$ photoreduction reactions. However, this process requires multiple setups, and there is no standard strategy to separate and analyze all these products, reactants, and air components simultaneously. Even for the most reported photocatalytic $CO_2$ conversion process of $CO_2$ to CO, a mixture of $O_2$, $N_2$, CO, $CH_4$, $CO_2$, and water vapor cannot be completely separated with one column due to the irreversible adsorption of $CO_2$ in the HP-Molesieve column and poor separation of $CO/N_2$ in the HP-PLOT/Q column. To solve this problem, we designed a parallel connection system (Fig. 5a, see "Methods" for experimental details) containing both HP-Molesieve and HP-PLOT/Q columns. The diameters of the parallel GC columns were optimized to enable the components from different columns to queue into the MS detector separately. As shown in Fig. 5b, $O_2$ (RT at 1.79 min), $N_2$ (RT at 2.20 min), $CH_4$ (RT at 2.48 min), and CO (RT at 4.59 min) elute from HP-Molesieve one by one and generate the corresponding MS spectra (Fig. 5c). It matches the abovementioned standard MS in Fig. 4a, respectively. $CO_2$ and $H_2O$ could not be resolved by the HP-Molesieve channel but could be measured in the parallel HP-PLOT/Q channel. They eluted in the HP-PLOT/Q channel as an $N_2/O_2/CO$ mixture (RT at 10.68 min), $CH_4$ (RT at 11.10 min), $CO_2$ (RT at 12.03 min), and $H_2O$ (RT at 16.91 min), which matches the standard MS in Fig. 4b. Based on this parallel strategy, the simulated mixture from reagents to products to air components in the potential photocatalytic $CO_2$ reduction process could be analyzed in one GC-MS experiment.

**Testing the isotope-tracing protocol using samples generated by well-studied $CO_2$ photoreduction reactions**

After optimizing our isotope-tracing methods using multiple standard samples, we tried to verify its reliability using some previously reported photoreduction systems. The homogeneous molecular catalyst $Fe^{III}$ porphyrin complex was used to evaluate $CO_2$ photoreduction (see "Methods" for experimental details) in a closed gas circulation system (Supplementary Fig. 21)[59]. As shown in Fig. 6a, the obtained TIC of this

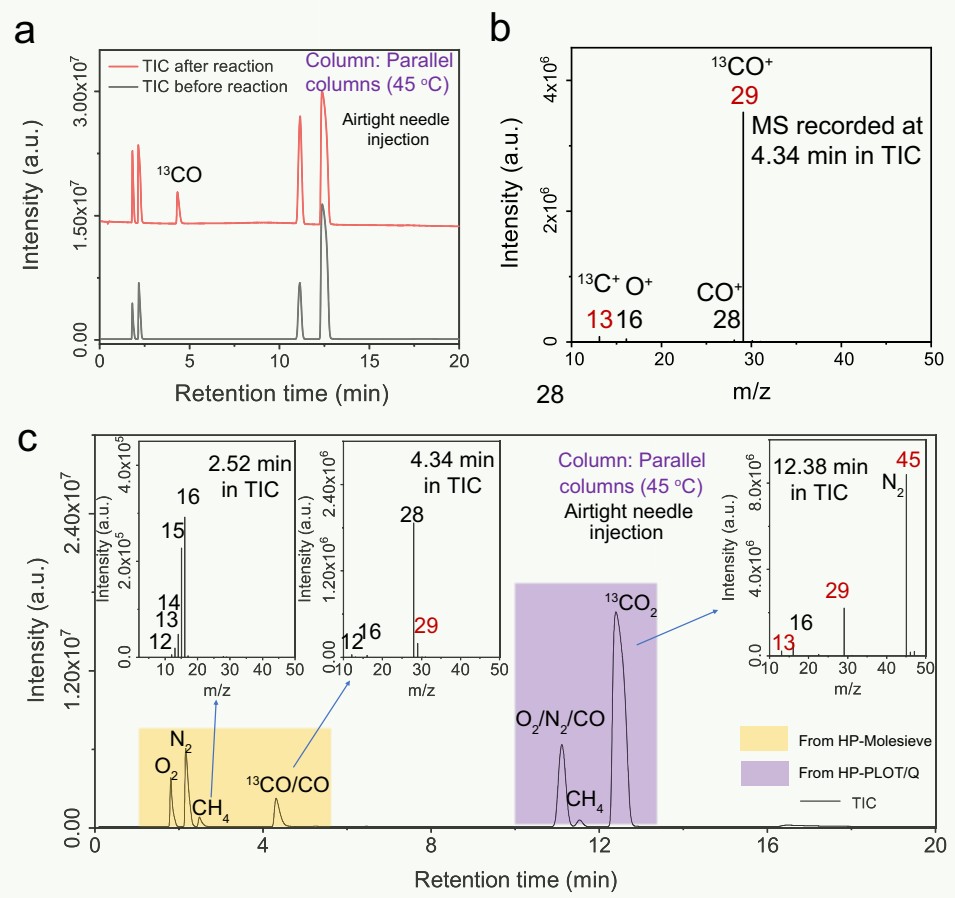

**Fig. 7 | Isotope-tracing experiments in reported liquid and gas–solid phase photoreduction systems.** GC-MS analytical data for $^{13}C$ isotope-tracing experiments in $CO_2$ photoreduction using the following catalysts: conjugated polymers in liquid (**a**, **b**) and gas–solid phase (**c**). Each panel shows the TICs and MS spectra of products generated by the catalysts in a $CO_2$ photoreduction reaction. Source data are provided as a Source Data file.

system is similar to the standard spectra (Fig. 4a) because $CH_4$ and CO are products of this reaction. In MS, the molecular ion peaks, as well as the fragment ion peaks, were assigned to the isotopically labeled $^{13}CH_4$ and $^{13}CO$, respectively, conclusively demonstrating that the products of $CH_4$ and CO originated from the photoreduction of labeled $^{13}CO_2$ (Supplementary Fig. 22). Bipyridine complexes of ruthenium ($Ru(bpy)_3Cl_2$) is also a common homogeneous catalyst for $CO_2$ photoreduction. By comparison with the sample without reaction (Supplementary Fig. 23), we could confirm that the $^{13}CO$ product comes from the reduction of $CO_2$ but not from the interfering factor of reactants (fragment ions from $^{13}CO_2$) (see "Methods" for experimental details). We also examined $CO_2$ photoreduction using a heterogeneous reaction system with Cd-doped ZnS as the photocatalyst. Cd-doped ZnS can reduce $CO_2$ into HCOOH under light irradiation and was used to validate our GC-MS isotope-tracing method for liquid-phase products (see "Methods" for experimental details)[60]. Fig. 6 shows the TIC (Fig. 6c) and MS spectrum (Fig. 6d) collected using the products. The peak at RT = 10 min in the TIC corresponded to three peaks in the MS that match our standard MS for $H^{13}COOH$ (Fig. 4b). These results demonstrate that our method is also capable of tracing liquid products generated by $CO_2$ reduction.

In addition, isotope tracing in all-organic reaction systems is particularly difficult to assess in $CO_2$ photoreduction experiments. We employed our recently reported conjugated polymers (CPs) as a representative photocatalyst to experiment with both gas–solid phases (Supplementary Fig. 24) (see "Methods" for experimental details) and liquid phase (Supplementary Fig. 25) (see "Methods" for experimental details) reactions[34]. Only an increase in CO could be observed

during the liquid phase reaction by comparing the TIC before and after photoreduction (Fig. 7a), and the MS collected at RT = 4.59 min matches the standard MS of $^{13}CO$ (Fig. 7b), implying that CO originates from $CO_2$ reduction (Supplementary Fig. 26). In contrast, CPs in the gas phase reaction possesses much higher activities and produces additional $CH_4$ compared to the liquid phase reaction. In the corresponding MS (inset of Fig. 6c), CO and $CH_4$ are primarily generated by photothermal effect-induced decomposition on solid CP instead of $CO_2$ photoreduction (Supplementary Fig. 27). This result leaves little doubt that the self-decomposition of catalyst can seriously affect both the qualitative and quantitative analysis of the product when evaluating the $CO_2$ photoreduction in the gas-solid phase of CPs. It also illustrates the importance of accurate isotope-tracing methods to prove the efficiency of $CO_2$ photoreduction in different materials systems. Moreover, it should be emphasized that the isotope experiments and accurate GC-MS tests are not only necessary for the reaction systems using sacrificial agents but also important for the reaction systems without sacrificial agents.

## Discussion

In summary, we initially demonstrated the difficulty of accurately ascribing products of $CO_2$ photoreduction in isotope-tracing experiments. Existing methods are known to be rather crude due to the similarity of reactants, products, and the catalyst itself, which negatively affects efforts toward reliable $CO_2$ photoreduction processes. Thus we sought to develop a rigorous strategy to eliminate the false-positive results by providing solid evidence of $CO_2$ photoreduction. We disclosed many often neglected false-positive results in isotope-tracing

studies using standard isotopically labeled molecules and the basic principles of GC-MS analysis. Through extensive testing, we systematically presented isotope-tracer experiments methods of standard spectra for various potential products of CO, alkanes, alcohols, carboxylic acids, and alkenes. Then we devised a method to simultaneously analyze these potential products using a parallel connection system using two GC columns. The accuracy of this parallel setup was verified with the standard isotopes described earlier; then, the setup was used to analyze four previously reported systems for $CO_2$ photoreduction. The purpose of this research is to highlight appropriate scientific methods in $CO_2$ photoreduction, help researchers avoid the pitfalls and misunderstandings in isotope-tracing experiments, and propose some standard procedures. More precise procedures will allow researchers to provide better feedback in their efforts to design photocatalysts with high selectivities and conversion efficiencies for $CO_2$ photoreduction. We also provide examples and reference measurements for the $^{13}C$ isotopes of numerous known reactants and products to help researchers firmly attribute them to $CO_2$ reduction reactions.

## Methods

### Using HP-5ms as a column for isotope-labeled samples analysis
The 0.5 ml gas samples were collected and injected by gas-tight syringes (the VICI Pressure-Lok Precision Analytical Syringe A-2 Series (050033), 1 ml) and then analyzed by gas chromatography-mass spectrometry (8890-5977B GC-MS instrument, Agilent Technologies, USA) equipped with most used commercial capillary columns (HP-5ms, 5%-Phenyl-methylpolysiloxane, 19091S-433UI-KEY, 30 m × 0.25 mm × 25 μm, Agilent Technologies, USA) in GC-MS. Helium was used as carrier gas. The column was maintained at 150 °C for 25 min, and the flow of the carrier was 0.8 ml l$^{-1}$. The temperatures of the injector, EI source, and GCITF were set to be 200, 200, and 250 °C, respectively. The selected mass-to-charge ratios of ions were 17, 29, and 45 in SIM mode. Developing a suitable programmed temperature rise process can further shorten the detection time.

### Using SVUV-PIMS for isotope-labeled samples analysis
The synchrotron VUV photoionization mass spectroscopy (SVUV-PIMS) experiments were performed on the combustion station of the National Synchrotron Radiation Laboratory (Hefei, China). The test modes are the photoionization energy scan under the determined mass-to-charge ratio and the mass-to-charge ratio scan under the determined photoionization energy, respectively. Synchrotron radiation from the undulator beamline was monochromatized with 200 lines/mm laminar grating (Horiba Jobin Yvon, France), which covered the photon energy from 7.5 to 22 eV with an energy resolving power of 3000 (E/ΔE at 10 eV). The average photon flux could reach the magnitude of 1013 photons/s after suppressing the higher-order harmonic radiation by a gas filter filled with noble gas[61,62].

### Using different columns for isotope-labeled samples analysis
The 0.5 ml gas samples were collected and injected by gas-tight syringes (the VICI Pressure-Lok Precision Analytical Syringe A-2 Series (050033), 1 ml), and the 2 ml liquid samples were collected and placed in a headspace sampler; then the samples were analyzed by gas chromatography-mass spectrometry (8890-5977B GC-MS instrument, Agilent Technologies, USA) equipped with commercial capillary columns. The column was maintained at a certain temperature for 15 min, and the flow of the carrier was 0.8 ml l$^{-1}$. The temperatures of the injector, EI source, and GCITF were set to be 200, 200, and 250 °C, respectively. The mass-to-charge ratio of the mass scanning mode was set from 2 to 70. The GC-MS was operated the post-run after each injection (the temperature of the column oven increased to 300 °C with a rate of 30 °C and then maintained at 300 °C for 10 min). Developing a suitable programmed temperature rise process can further shorten the detection time.

The information of the columns are listed below:
(HP-Molesieve, 5A molesieve, 19091S-MS8, 30 m × 0.32 mm × 25 μm, Agilent Technologies, USA; HP-PLOT/Q, Bonded polystyrene-divinylbenzene, 19091P-QO4, 30 m × 0.32 mm × 20 μm, Agilent Technologies, USA; HP-FFAP, Modified polyethylene glycol, 19091F-413, 30 m × 0.32 mm × 20 μm, Agilent Technologies, USA).

### Using a designed parallel connection system as a column for isotope-labeled samples analysis
The 0.5 ml gas samples were collected and injected by gas-tight syringes (the VICI Pressure-Lok Precision Analytical Syringe A-2 Series (050033), 1 ml) and then analyzed by gas chromatography-mass spectrometry (8890-5977B GC-MS instrument, Agilent Technologies, USA) equipped with designed parallel connection system (HP-Molesieve 15 m × 0.53 mm × 20 μm, HP-PLOT/Q 15 m × 0.32 mm × 20 μm, and CP4016 10 m × 0.32 mm, Agilent Technologies, USA) in GC-MS. Helium was used as carrier gas. The column was maintained at 45 °C for 20 min, and the flow of the carrier was 0.8 ml l$^{-1}$. The temperatures of the injector, EI source, and GCITF were set to be 200, 200, and 250 °C, respectively. The mass-to-charge ratio of the mass scanning mode was set from 2 to 70. The GC-MS was operated the post-run after each injection (the temperature of the column oven increased to 300 °C with a rate of 30 °C and then maintained at 300 °C for 10 min). Developing a suitable programmed temperature rise process can further shorten the detection time.

## Data availability
The data that support the plots within this paper and other findings of this study are available from the corresponding author upon reasonable request. Source data are provided with this paper.

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

## Acknowledgements

This work received financial support from the National Science Foundation of China (Nos. 51902121, 52073110, 22106106, 22071072, 21975090, and 52003213), the World Premier International Research Center Initiative (WPI Initiative) on Materials Nanoarchitectonics (MANA), MEXT (Japan), KAKENHI (18H02065, 20K05453), MEXT, and Photoexcitonix Project in Hokkaido University, Japan, the Fundamental Research Funds for the Central Universities of China (2662022LXPY001, 2662018QD011, 2662018PY052, SZYJY2022012, and 2662019PY023), and the Natural Science Foundation of Hubei Province (2019CFB322).

## Author contributions

S.W. and J.Ye. conceived the idea and designed the experiments; S.W. established methods for GC-MS analysis; S.W. and B.J. designed the parallel connection system; S.W., H.S., and S.Z. conducted GC-MS analysis for isotope-labeled standard samples; F.X., C.L., and Y.P. conducted the SVUV-PIMS measurements; S.W., X.M., and B.J. performed the isotopic experiments over reported systems. S.W., J.H., and B.J. wrote the manuscript; H.L., J.Yu., H.C., and J.Ye. jointly supervised the work and revised the manuscript. All authors contributed to the discussion of the results and the manuscript preparation.

## Competing interests

The authors declare no competing interests.
