## [Peer Review File · Nature Communications]

Designing Reliable and Accurate Isotope-Tracer Experiments in CO₂ PhotoreductionREVIEWER COMMENTS

Reviewer #1 (Remarks to the Author):

This manuscript describes standard guidelines for isotope tracing experiments in CO₂ photoreduction reactions and then verify the procedure to using some reported photoreduction systems. The greatest contribution of this work is the explanation of the mass spectrometry data, with an almost complete analysis and comparison of all signal sources. However, in the use of chromatography, the authors seem not to have fully investigated the current commercial equipment and technical solutions. In summary, although the current commercial GC-MS equipment is well equipped for product separation and detection, it is still a meaningful effort to analyze the mass spectrometry data in such detail. Thus, it is recommended to publish in Nat. Commun. upon revisions.

1. The authors used a series of standard pure gases for mixing to obtain a gas mixture (without H₂) that simulates the CO₂ reduction product. However, in real reactions, H₂ is often the dominant competing product. The authors chose helium as the carrier gas when using chromatography for the separation. Although this is feasible within the scope of this work, it can be unfavorable for the separation or detection of H₂ in chromatography when faced with a practical situation. Have the authors considered the influence of carrier gas on the conclusions of this work?
2. The authors designed a parallel connection system (Figure 5a) to achieve the separation and detection of the gas phase products of CO₂ reduction. Among them, a HP-Molesieve column was used to separate O₂, N₂, etc. It is worth noting that CO₂ gas can poison the HP-Molesieve column. However, I have not seen any information about the methods to avoid the flow of large amounts of CO₂ into the HP-Molesieve column (back-flushing, etc.). The authors should give more experimental details.
3. In all tests, there is always the presence of an air component. Have the authors considered using a gas sample valve (e.g., 10-port Valve)? What are the advantages and disadvantages of using a syringe injection? If air is present in the sample, would it imply that there is a leakage of the reduction products?
4. The current method may require nearly 20 minutes of analysis time, have the authors considered using a programmed temperature rise system?
5. As I have previously stated, GC technology is mature enough to allow for excellent separation of multiple components. The authors spend a great effort in describing and optimizing the choice of columns. Have the authors compared with current commercial instruments? What are the advantages and disadvantages of the system designed by the authors?

Reviewer #2 (Remarks to the Author):

In their submitted article, Shengyao Wang et al. are testing various equipment and procedures in order to establish a reliable ¹³C isotope labelling strategy for product verification in CO₂ reduction. They unveil several pitfalls that lead to false-positive results, and then propose a strategy/equipment configuration to exclude such false results. This strategy is then used to test a number of homogeneous and heterogeneous CO₂ reduction photocatalysts and verify or falsify product formation from CO₂.

The intention of the authors is very positively acknowledged. As is properly stated in the introduction, the results in the whole research field are often doubted by the research community, because we still have not learned to fully exclude false positive results. Thus, strategies as the one presented by the author are dearly needed. All in all, the paper is well presented, so that publication in Nature Communications is very much desired. However, a major revision is necessary to address the issues outlined below:

(1) The introduction properly cites Asian and North American works on true product identification in CO₂ reduction, but lacks citations of early works from Europe, in particular the works from Guido Mul (C.-C. Yang, Y.H. Yu, B. van der Linden, J.C.S. Wu, G. Mul, *J. Am. Chem. Soc.* 132 (2010) 8398; C.-C. Yang, J. Vernimmen, V. Meynen, P. Cool, G. Mul, *J. Catal.* 284 (2011) 1.), Adriana Zaleska (A. Cybula, M. Klein, A. Zaleska, *Appl. Catal. B Environmental* 164 (2015) 433.) and the works from my own group on CO₂ reduction under high-purity reaction conditions (for example, B. Mei, A. Pougin, J. Strunk, *J. Catal.* 306 (2013) 184; N.G. Moustakas, J. Strunk, *Chem. Eur. J.* 24 (2018) 12739.). In order to give a complete historical overview, these works should be added, even more so since Guido Mul also uses ¹³C labelling, and we also described the GC system in much detail (Mei et al.).

(2) The analysis of the false positive results with the wrong column or wrong detection method (SIM) is interesting, but as these results are presented now, one could argue that they are little relevant, because obviously unsuitable equipment will naturally lead to "bad" results. My recommendation is to extend this paragraph slightly, in order to discuss that this would also be the outcome with pure MS detection (without GC) where all components enter the quadrupole together. This is a much more common case in recent literature, so it would make the paragraph more generally relevant.

(3) The authors write in line 319 "the full photocatalytic CO₂ reduction process". This seems a little overestimated, because it is still a "simplified" mixture, in which CO, CH₄, O₂ (as impurity or byproduct), N₂, CO₂ and H₂O are considered. The whole issues would be much more complicated if also higher hydrocarbons, alcohols and acids would be considered. It is suggested to change the sentence to a more careful version.

(4) From the experimental description in the SI, it does not become clear if a sacrificial reagent has been added to the homogeneous photocatalysts. Furthermore, more information is needed on the reaction conditions for testing the CP samples.

(5) If sacrificial reagents have been added in all liquid phase reactions, it is quite remarkable that true product formation has only been observed with sacrificial reagents, but not in the gas-solid reaction. Can the authors use commercial TiO₂ P25? As we (and others) have shown, P25 shows CH₄ formation under high-purity gas-solid conditions without sacrificial reagent, so it would be worthwhile to confirm (or falsify!) this result by the authors' new isotope-labelling procedure.

(6) A few minor issues also need the attention of the authors:

(i) Title should be changed. Either use "a reliable [...] tracer experiment" (singular) or "reliable [...] tracer experiments" (plural, without "a"); (ii) line 100: Mass fragments should also be mentioned (i.e., "and an MS to detect their mass, or their mass fragments."); (iii) Synthesis of Cd-doped ZnS: Cd source is not mentioned; (iv) In all experimental procedures, the purity of helium should be provided. For such highly accurate procedures, this is a very relevant information; (v) It is suggested to shorten the experimental description of the different column testing. It appears to be always the same, except for the column type and temperature. So, it is suggested to describe it only once, and in addition to provide a Table with a list of the tested columns and their temperatures. (vi) In Figure S21b and the Caption of Figure S26 (line 404) it should read "CO" not "CO₂".

Thank you for the nice work, and all the best for the revision,
Jennifer Strunk

Reviewer #3 (Remarks to the Author):

This paper addresses one major concern regarding photochemical CO₂ reduction : the origin of products. Are they really coming from CO₂ or from other sources ? This apparently simple question relies on the proper implementation of labelled studies (using ¹³CO₂ to detect ¹³-labelled products), which may be achieved by various techniques, notably upon coupling GC (gas chromatography) to MS detection (mass spectrometry). NMR (¹H) spectroscopy may also be used and I will comment on that later in the review. GC-MS technique, to be reliable, needs to combine excellent separation of products through the GC analysis, and then proper identification of each individual product by MS analysis, which necessitates to also have fragmentation pattern being identical to authentic sample. Having said that, it is clear that a significant fraction of current literature does not fulfill these criteria (including in high level journals, including in recent results), which significantly slows down progress in the field and also hampers the credibility of our community.

The authors have designed an interesting and rigorous experimental protocol to identify CO₂ photoreduction products by GC-MS. They have further set conditions and described results for some of the main targeted products such as CO, CH₄, methanol and ethanol. Finally, taking examples from literature, they have applied their methods to previous systems, identifying drawbacks and sometimes incorrect results from these previous works.

Overall the paper is technically well executed and provides many insights and guidelines for people working in the field. It participate to the necessary collective effort to have better procedures to assess the quality and the reality of the results.

There is one significant drawback in this work : the authors never mentioned an alternative method,

which is much more simple than the two lines GC-MS and that is, contrary to the author statement (lines 301-304), perfectly suited for a one single, simple experiment (even an undergraduate student can do it properly and accurately) separation, identification and quantification of all components simultaneously. This method is ^1H NMR. Of course, it is pertinent for liquid products, so that it should be viewed as a complementary technique to GC-MS. But if one consider a mixture of e.g. methanol, ethanol, formate/formic acid, acetic acid, one single spectra under $^{12}\text{CO}_2$ and then under $^{13}\text{CO}_2$ can demonstrate the origin of the products, and the exact quantity of these products, with NO interference of any kind between ALL these products. For example, authors may consult with profit Dalton Trans. 2020, 49, 4257-4265, that illustrates the simplicity and high efficiency of this method, as well as papers from the Jaramillo group in Stanford. There is NMR equipment in all labs, which is another huge advantage of such an approach. Moreover, crossing results from several techniques is also extremely important when assessing the product origin in CO_2 reduction. So I believe that the authors should clearly mention this approach and its remarkable efficiency for liquid product analysis, even through one single spectra, even in case of complex mixtures, where identification through peak shifts and coupling constants has already been tabulated for barely all C_1 to C_3 products. Complementarity between techniques should also be clearly highlighted.

Overall, such modifications will strengthen the paper and should be implemented. I would be happy to quickly assess a revised version of the paper.

Reviewer #4 (Remarks to the Author):

General comment

The author has made a sincere effort in this work to address the vital topic of the CO_2 photoreduction reaction. The author examined the scientific and challenging feature of the isotopic test in CO_2 reduction that has to be clarified to avoid misunderstanding. Consequently, they have effectively designed an experimental arrangement to deliver significantly greater reliability than the current experimental approach. However, much additional analysis and revisions are necessary. The fundamental problem is that the existing explanation in the paper lacks impact or importance concerning the topic. As a result, the authors must carefully handle this issue, and additional comments are provided below.

Comments

1. Several research in the literature demonstrate the applicability of SVUV-PIMS for CO_2 reduction; we recommend that the author discuss these studies in the introduction. A representative research paper is provided below.

Shao, Weiwei, et al. "In-plane heterostructure Ag_2S - In_2S_3 atomic layers enable boosted CO_2 photoreduction into CH_4 ." Nano Research 14.12 (2021): 4520-4527.

2. The author discussed the combination of $^{13}\text{CO}_2$, ^{13}CO , and water vapour. Is there a particular

rationale for selecting $^{13}\text{CO} > 5\%$ v/v? Additionally, if we alter the combination, v/v will affect the results; whether yes or no, please provide the pertinent information.

3. The author claims that the formation of HO^+ and $^{13}\text{CO}^+$ raises questions on the study; is it possible to provide additional proof for this claim? In addition, HO^+ must, if possible, demonstrate the H_2O isotopic test.

4. Several research on SVUV-PIMS have previously been published; thus, it is understandable that the author wants this analysis to investigate this technique. Nonetheless, it appears to be a typical analysis. We believe the significance of this analysis merits further investigation. This approach and existing modifications are missing connections. Thus, we suggest that the author must have coherence.

5. There is considerable concern that HP-PLOT Molesieve Columns are commercially accessible; we appreciate the concept of putting HP-PLOT and Molesieve in parallel. Nevertheless, the author can elaborate on how this methodology will contribute to scientific importance.

6. Figure S21a photocatalytic system still shows the $^{16}\text{CH}_4$, is this surface contamination of photocatalyst? In Fe(III) porphyrin complex, while synthesising, all the precursors are organic, and also, there is no high-temperature treatment; how does the author avoid carbon contamination?

7. In the case of the photocatalytic activity, it is time-dependent; for instance, is it possible to show the time-dependent GC-MS? In Figure S22, with light and without light radiation, how do we understand that the concentration of $^{13}\text{CO}_2$ changes and converts to ^{13}CO ?

8. In the gas phase, we strongly suggest the added some result and discussion about the time-dependent study of CO_2 photoreduction with GC-MS analysis.

9. In the case of liquid-phase CO_2 photoreduction, several photocatalyst systems employ organic solvents and sacrificial reagents containing organic moieties (Such as TEOS). Does the author believe that this will impede the separation of the product? Please include some experiments and an explanation, if feasible.

10. In many figures, captions are not correctly mentioned. Hence it is difficult to refer to the corresponding figure; please check the supporting information in the caption; there is no a), b) and so on mentioned.

11. I recommend for the authors to study recently published papers related to this work.

- "Solar fuels: Research and development strategies to accelerate photocatalytic CO_2 conversion into hydrocarbon fuels", Energy & Environmental Science 15 (2022) 880 - 937

- "Electronic interaction between transition metal single-atoms and anatase TiO_2 boosts CO_2 photoreduction with H_2O ", Energy & Environmental Science 15 (2022) 601-609

Point-by-point response to the reviewers' comments.

Reviewer #1 (Remarks to the Author):

This manuscript describes standard guidelines for isotope tracing experiments in CO₂ photoreduction reactions and then verify the procedure to using some reported photoreduction systems. The greatest contribution of this work is the explanation of the mass spectrometry data, with an almost complete analysis and comparison of all signal sources. However, in the use of chromatography, the authors seem not to have fully investigated the current commercial equipment and technical solutions. In summary, although the current commercial GC-MS equipment is well equipped for product separation and detection, it is still a meaningful effort to analyze the mass spectrometry data in such detail. Thus, it is recommended to publish in Nat. Commun. upon revisions.

Response: Thank you very much for your kind and valuable comments. Your suggestions are helpful in improving the quality of the manuscript.

1. The authors used a series of standard pure gases for mixing to obtain a gas mixture (without H₂) that simulates the CO₂ reduction product. However, in real reactions, H₂ is often the dominant competing product. The authors chose helium as the carrier gas when using chromatography for the separation. Although this is feasible within the scope of this work, it can be unfavorable for the separation or detection of H₂ in chromatography when faced with a practical situation. Have the authors considered the influence of carrier gas on the conclusions of this work?

Response: Thank you very much for your comment. Indeed, H₂ is often the dominant competing product in the real reaction of photocatalysis. In fact, we have considered the gas mixture with H₂ to simulate the CO₂ photoreduction product. The detection of H₂ (molecular ion peaks: H₂⁺, m/z=2, and fragment ion peaks H⁺, m/z=1) in mass spectrometry (MS) will be influenced by the lower limit of the mass range (m/z>1) for quadrupole mass analyzer and the molecular ion peaks of He⁺, m/z=2. When choosing He as the carrier gas, we can detect hydrogen (H₂ and D₂) (Figure R1). But we found that H₂ and D₂ are easily ionized and formed water (H₂O and D₂O) with residual trace oxygen (an environment of absolute vacuum that cannot be achieved on the earth) of channel in GC-MS. However, the MS of D₂O (D₂O⁺, m/z=20; DO⁺, m/z=18; O⁺, m/z=16) and H₂O (H₂O⁺, m/z=18; DO⁺, m/z=17; O⁺, m/z=16) will interfere with each other (Figure R2). So, it is very difficult to achieve the detection of hydrogen when the carrier gas is He.

Figure R1. The TIC spectra of hydrogen (H_2 and D_2) and Air using high purity helium as the carrier gas for GC-MS.

Figure R2. The mass spectra of hydrogen (H_2 and D_2) that recorded at 2.85 min in TIC.

Of course, according to your suggestion, other carrier gas (H_2 , N_2 , and Ar) were also considered in GC-MS technology to confirm the influence of carrier gas on the conclusions. Owing to the ionization energy of N_2 (15.6 eV), it is easily ionized in MS, which could have

relatively large background interference. We cannot carry out the detection of H₂ under this condition. For the carrier gas of H₂, it will undoubtedly interfere with the detection of hydrogen. When choosing Ar as the carrier gas, we can also detect the hydrogen (H₂ and D₂) (Figure R3). Based on the results, the ionized Ar⁺ can react with H₂ and D₂ to form ArH⁺ (m/z= 41) and ArD⁺ (m/z= 42) that could be measured in the MS spectra (J. Chem. Phys. 1985, 83, 166). While this potential pathway for the detection of H₂ in MS still needs to be developed and improved. Developing isotope traceability of hydrogen is essential but challenging, and we have devoted our time to the relevant research works. We hope to show the detailed results in our subsequent paper.

Figure R3. The TIC spectra of hydrogen (H₂ and D₂) and Air using high purity argon as the carrier gas for GC-MS (a) and the corresponding the mass spectra (b) that recorded at 3.37 min in TIC.

2. The authors designed a parallel connection system (Figure 5a) to achieve the separation and detection of the gas phase products of CO₂ reduction. Among them, a HP-Molesieve column was used to separate O₂, N₂, etc. It is worth noting that CO₂ gas can poison the HP-Molesieve column. However, I have not seen any information about the methods to avoid the flow of large amounts of CO₂ into the HP-Molesieve column (back-flushing, etc.). The authors should give more experimental details.

Response: Thank you very much for your suggestion. The HP-Molesieve column is used to separate H₂, O₂, N₂, and CO₂ can poison the HP-Molesieve column. Back-flushing usually acts as the method in GC to avoid the flow of large amounts of CO₂ into the HP-Molesieve column. But for GC-MS, the requirement for vacuum levels makes the back-flushing a little bit difficult and increases the costs. Although the pressure-controlled tee device (PCT) could operate the back-flushing method, it is an unavoidable influence for detecting C₂ alkanes. When a parallel connection system is used as the column, loading back-flushing becomes much more difficult. On the other hand, we only injected 0.5-1 mL gases each time and most of the CO₂ in HP-Molesieve column can be desorbed during the heating process of post-run.

We can find that the toxicity of CO₂ for the HP-Molesieve column is relatively weak and the separation efficiency of the column remained good after hundreds of use.

3. In all tests, there is always the presence of an air component. Have the authors considered using a gas sample valve (e.g., 10-port Valve)? What are the advantages and disadvantages of using a syringe injection? If air is present in the sample, would it imply that there is a leakage of the reduction products?

Response: Thank you very much for your comment. Similar to back-flushing, 10-port Valve is also a useful injection method for gas samples in GC detection. When applied to GC-MS, the low flow rate will affect the process of injection by 10-port Valve, which may result in an offset in the peak position during the MS detection. Although there are certain deficiencies in accurately controlling the injection volume for the syringe injection, it is a much easier way to inject gas samples for qualitative isotope detection. As the reviewer mentioned, air is present in almost all samples. Actually, we excluded the possibility of the leakage in the present system, since the changes in the amount of ¹³CO₂ or ¹³CO is negligible after sealing a long time. Instead, the trace oxygen that can be detected because an absolute vacuum cannot be achieved in any current vacuum condition. We are devoting our efforts to developing an online system with weak air interference for the difficult ¹³C quantitative detection; we would like to introduce this advanced system in our subsequent paper.

4. The current method may require nearly 20 minutes of analysis time, have the authors considered using a programmed temperature rise system?

Response: Thank you very much for your valuable suggestion. We agree with your idea that a programmed temperature rise system is much more suitable for the testing of authentic sample. We can even modify the programmed temperature rise system for each different sample. In this article, we are trying to confirm that the current discussion on isotope tracing experiments in CO₂ photoreduction is not rigorous. So, we didn't put much effort into developing various programmed temperature rise processes but chose the most uncomplicated constant temperature process. Considering the reviewer's suggestion, we added the following sentence in the Chromatographic method sections **"Developing a suitable programmed temperature rise process can further shorten the detection time"**

5. As I have previously stated, GC technology is mature enough to allow for excellent separation of multiple components. The authors spend a great effort in describing and optimizing the choice of columns. Have the authors compared with current commercial instruments? What are the advantages and disadvantages of the system designed by the authors?

Response: Thank you very much for your comment. We strongly agree that the GC technology is mature enough to allow for excellent separation of multiple components. However, all these commercial instruments rely on multi-valve for flow path switching and the coordination multi-column, which are unsuitable for GC-MS systems due to the pressure of the flow path decreasing. As far as we know, the current commercial instruments for GC

cannot be used in GC-MS directly, and there is no mature strategy for GC-MS to allow for excellent separation of multiple components. Such as, a mixture of O₂, N₂, CO, CH₄, CO₂, and water vapor cannot be separated with one column due to the irreversible adsorption of CO₂ in the HP-Molesieve column and poor separation of CO/N₂ in the HP-PLOT/Q column. To solve this problem, we designed a parallel connection system containing both HP-Molesieve and HP-PLOT/Q columns. Based on this parallel strategy, the photocatalytic CO₂ reduction process from reagents to products to air components could be analyzed in GC-MS experiment for the first time.

Reviewer #2 (Remarks to the Author):

In their submitted article, Shengyao Wang et al. are testing various equipment and procedures in order to establish a reliable ¹³C isotope labelling strategy for product verification in CO₂ reduction. They unveil several pitfalls that lead to false-positive results, and then propose a strategy/equipment configuration to exclude such false results. This strategy is then used to test a number of homogeneous and heterogeneous CO₂ reduction photocatalysts and verify or falsify product formation from CO₂.

The intention of the authors is very positively acknowledged. As is properly stated in the introduction, the results in the whole research field are often doubted by the research community, because we still have not learned to fully exclude false positive results. Thus, strategies as the one presented by the author are dearly needed. All in all, the paper is well presented, so that publication in Nature Communications is very much desired. However, a major revision is necessary to address the issues outlined below:

Response: Thank you very much for your kind and valuable comments. Your suggestions are helpful in improving the quality of the manuscript.

(1) The introduction properly cites Asian and North American works on true product identification in CO₂ reduction, but lacks citations of early works from Europe, in particular the works from Guido Mul (C.-C. Yang, Y.H. Yu, B. van der Linden, J.C.S. Wu, G. Mul, J. Am. Chem. Soc. 132 (2010) 8398; C.-C. Yang, J. Vernimmen, V. Meynen, P. Cool, G. Mul, J. Catal. 284 (2011) 1.), Adriana Zaleska (A. Cybula, M. Klein, A. Zaleska, Appl. Catal. B Environmental 164 (2015) 433.) and the works from my own group on CO₂ reduction under high-purity reaction conditions (for example, B. Mei, A. Pougin, J. Strunk, J. Catal. 306 (2013) 184; N.G. Moustakas, J. Strunk, Chem. Eur. J. 24 (2018) 12739.). In order to give a complete historical overview, these works should be added, even more so since Guido Mul also uses ¹³C labelling, and we also described the GC system in much detail (Mei et al.).

Response: Thanks a lot for the suggestion. We have added these references in the revised manuscript.

15 Pougin, A. et al. Au@TiO₂ core-shell composites for the photocatalytic reduction of CO₂. *Chemistry-A European Journal* **24**, 12416-12425 (2018).

16 Dulla, M., Pougin, A. & Strunk, J. Evaluation of the plasmonic effect of Au and Ag on Ti-based photocatalysts in the reduction of CO₂ to CH₄. *Journal of Energy Chemistry*

- 26, 277-283 (2017).
- 17 Moustakas, N. G., Strunk, J. Photocatalytic CO₂ reduction on TiO₂-based materials under controlled reaction conditions: systematic insights from a literature study. *Chemistry-A European Journal* **24**, 12739-12746 (2018).
- 26 Yang, C. C., Vernimmen J., Meynen, V., Cool, P. & Mul, G. Mechanistic study of hydrocarbon formation in photocatalytic CO₂ reduction over Ti-SBA-15. *Journal of Catalysis* **284**, 1-8 (2011).
- 27 Mei, B., Pougin, A. & Strunk, J. Influence of photodeposited gold nanoparticles on the photocatalytic activity of titanate species in the reduction of CO₂ to hydrocarbons. *Journal of Catalysis* **306**, 184-189 (2013).
- 31 Yang, C. C., Yu Y. H., van der Linden, B., Wu, J. C. & Mul, G. Artificial photosynthesis over crystalline TiO₂-based catalysts: fact or fiction? *Journal of the American Chemical Society* **132**, 8398-8406 (2010).
- 32 Cybula, A., Klein, M. & Zaleska, A. Methane formation over TiO₂-based photocatalysts: reaction pathways. *Applied Catalysis B: Environmental* **164**, 433-442 (2015).

(2) The analysis of the false positive results with the wrong column or wrong detection method (SIM) is interesting, but as these results are presented now, one could argue that they are little relevant, because obviously unsuitable equipment will naturally lead to "bad" results. My recommendation is to extend this paragraph slightly, in order to discuss that this would also be the outcome with pure MS detection (without GC) where all components enter the quadrupole together. This is a much more common case in recent literature, so it would make the paragraph more generally relevant.

Response: Thank you very much for your valuable suggestion. In order to ensure the consistency of the injection process, we employed the deactivated fused silica tube (5m) without any separation effect as the connector to let all components (¹³CO₂, vapor, and air) enter the quadrupole together. As shown in **Figure R4**, except for a slightly different retention time in TIC, the prominent peak of all components can be found, which is similar to the results obtained from the system using the column of HP-5MS. This fully illustrates that the sample separation is critical for the MS detection.

Figure R4. The TIC spectra of the $^{13}\text{CO}_2$ and vapor mixture and the corresponding MS spectra recorded at 1.21 min in TIC with the deactivated fused silica tube (5m) as the connector.

We have added the following description in the revised manuscript.

"Even the deactivated fused silica tube (the length is 5m) without any separation effect act as the connector to let all components ($^{13}\text{CO}_2$ and vapor) enter the quadrupole together, a similar results can also be obtained in the mixture of $^{13}\text{CO}_2$ and vapor even without a photocatalytic reduction process (supplementary Fig. 4)."

(3) The authors write in line 319 "the full photocatalytic CO_2 reduction process". This seems a little overestimated, because it is still a "simplified" mixture, in which CO , CH_4 , O_2 (as impurity or byproduct), N_2 , CO_2 and H_2O are considered. The whole issues would be much more complicated if also higher hydrocarbons, alcohols and acids would be considered. It is suggested to change the sentence to a more careful version.

Response: Thank you very much for your suggestion. We have revised the following sentences in the current version.

"Based on this parallel strategy, the simulated mixture from reagents to products to air components in the potential photocatalytic CO_2 reduction process could be analyzed in one GC-MS experiment for the first time."

(4) From the experimental description in the SI, it does not become clear if a sacrificial reagent has been added to the homogeneous photocatalysts. Furthermore, more information is needed on the reaction conditions for testing the CP samples.

Response: Thank you very much for your comment. We added the following description in the revised supplementary information.

For the system of Fe^{III} porphyrin complex, the Fe^{III} porphyrin complex was dissolved in the acetonitrile/water solutions containing triethylamine as sacrificial electron donor, and $\text{Ir}(\text{ppy})_3$ were employed as sensitizer; For the system of Ru complex, the Ru complex was dissolved

in the acetonitrile/water solutions containing triethylamine as sacrificial electron donor; For the gas-solid system of CPs, CPs powders were uniformly dispersed onto a porous quartzose film which was fixed on the stage inside the reaction cell with addition of 3 mL of distilled water as electron donor; For the liquid system of CPs, CPs powders dispersed in the acetonitrile/water solutions containing triethylamine as sacrificial electron donor.

(5) If sacrificial reagents have been added in all liquid phase reactions, it is quite remarkable that true product formation has only been observed with sacrificial reagents, but not in the gas-solid reaction. Can the authors use commercial TiO₂ P25? As we (and others) have shown, P25 shows CH₄ formation under high-purity gas-solid conditions without sacrificial reagent, so it would be worthwhile to confirm (or falsify!) this result by the authors' new isotope-labelling procedure.

Response: Based on the suggestion from the reviewer, we employed the commercial TiO₂ (P25) as the photocatalyst to perform the CO₂ photoreduction under high-purity gas-solid conditions without sacrificial reagent. As shown in Figure R5, the CH₄ formation is gradually increased with the irradiation time, however, the MS spectra showed that the obtained CH₄ is not the ¹³C isotope labelled sample. It clearly demonstrates that carbon residues largely participate in the formation of CH₄ over P25 photocatalysts.

Figure R5 The TIC spectra of the sample collected from the CO₂ photoreduction system using P25 as photocatalyst at different irradiation time and the corresponding MS spectra recorded at 9.64 min in TIC.

(6) A few minor issues also need the attention of the authors:

(i) Title should be changed. Either use "a reliable [...] tracer experiment" (singular) or "reliable [...] tracer experiments" (plural, without "a"); (ii) line 100: Mass fragments should also be mentioned (i.e., "and an MS to detect their mass, or their mass fragments."); (iii) Synthesis of Cd-doped ZnS: Cd source is not mentioned; (iv) In all experimental procedures, the purity of helium should be provided. For such highly accurate procedures, this is a very relevant information; (v) It is suggested to shorten the experimental description of the different column

testing. It appears to be always the same, except for the column type and temperature. So, it is suggested to describe it only once, and in addition to provide a Table with a list of the tested columns and their temperatures. (vi) In Figure S21b and the Caption of Figure S26 (line 404) it should read "CO" not "CO₂".

Response: Thank you very much for your comment. We have revised (i) the title to "Realizing Reliable and Accurate Isotope-Tracer Experiments in CO₂ Photoreduction"; (ii) "The GC-MS instrument is composed of a GC to separate molecular species based on affinity with a column material and an MS to detect their mass, or their mass fragments."; (iii) "The Cd-doped ZnS can be obtained by adding a certain amount of Cadmium sulfate solution for ion exchange."; (iv) "The high purity helium (Chemical Purity 99.999%) with further purification by helium purifier (HP2, Valco Instruments Co. Inc) were used as the carrier gas for GC-MS."; (v) we have shortened the experimental description of the different column testing.; (vi) we have revised these errors.

Reviewer #3 (Remarks to the Author):

This paper addresses one major concern regarding photochemical CO₂ reduction: the origin of products. Are they really coming from CO₂ or from other sources? This apparently simple question relies on the proper implementation of labelled studies (using ¹³CO₂ to detect ¹³-labelled products), which may be achieved by various techniques, notably upon coupling GC (gas chromatography) to MS detection (mass spectrometry). NMR (¹H) spectroscopy may also be used and I will comment on that later in the review. GC-MS technique, to be reliable, needs to combine excellent separation of products through the GC analysis, and then proper identification of each individual product by MS analysis, which necessitates to also have fragmentation pattern being identical to authentic sample. Having said that, it is clear that a significant fraction of current literature does not fulfill these criteria (including in high level journals, including in recent results), which significantly slows down progress in the field and also hampers the credibility of our community.

The authors have designed an interesting and rigorous experimental protocol to identify CO₂ photoreduction products by GC-MS. They have further set conditions and described results for some of the main targeted products such as CO, CH₄, methanol and ethanol. Finally, taking examples from literature, they have applied their methods to previous systems, identifying drawbacks and sometimes incorrect results from these previous works. Overall the paper is technically well executed and provides many insights and guidelines for people working in the field. It participate to the necessary collective effort to have better procedures to assess the quality and the reality of the results.

Response: Thank you very much for your kind and valuable comments. Your suggestions are helpful in improving the quality of the manuscript.

There is one significant drawback in this work: the authors never mentioned an alternative method, which is much more simple than the two lines GC-MS and that is, contrary to the author statement (lines 301-304), perfectly suited for a one single, simple experiment (even

an undergraduate student can do it properly and accurately) separation, identification and quantification of all components simultaneously. This method is ^1H NMR. Of course, it is pertinent for liquid products, so that it should be viewed as a complementary technique to GC-MS. But if one consider a mixture of e.g. methanol, ethanol, formate/formic acid, acetic acid, one single spectra under $^{12}\text{CO}_2$ and then under $^{13}\text{CO}_2$ can demonstrate the origin of the products, and the exact quantity of these products, with NO interference of any kind between ALL these products. For example, authors may consult with profit Dalton Trans. 2020, 49, 4257-4265, that illustrates the simplicity and high efficiency of this method, as well as papers from the Jaramillo group in Stanford. There is NMR equipment in all labs, which is another huge advantage of such an approach. Moreover, crossing results from several techniques is also extremely important when assessing the product origin in CO_2 reduction. So I believe that the authors should clearly mention this approach and its remarkable efficiency for liquid product analysis, even through one single spectra, even in case of complex mixtures, where identification through peak shifts and coupling constants has already been tabulated for barely all C1 to C3 products. Complementarity between techniques should also be clearly highlighted.

Overall, such modifications will strengthen the paper and should be implemented. I would be happy to quickly assess a revised version of the paper.

Response: Thank you very much for your valuable suggestion. As mentioned by reviewer, the NMR is pertinent strategy to distinguish the isotope in liquid products. Initially, we have considered ^{13}C NMR spectra for the isotope-tracer study in CO_2 photoreduction, however, we found that ^{13}C NMR only corresponded to ^{13}C -labeled samples, the natural abundance of ^{13}C isotopes is detrimental to isotope detection. For the author mentioned ^1H NMR strategy, we have indeed ignored that it could achieve isotope identification by detecting the split phenomenon of ^{13}C -linked hydrogen in liquid sample. Consequently, we employed the non-isotope labeled standards (CH_3OH , $\text{CH}_3\text{CH}_2\text{OH}$, HCOOH and CH_3COOH) and the corresponding ^{13}C -labeled isotopic standards ($^{13}\text{CH}_3\text{OH}$, $\text{CH}_3^{13}\text{CH}_2\text{OH}$, $^{13}\text{CH}_3^{13}\text{CH}_2\text{OH}$, H^{13}COOH , $\text{CH}_3^{13}\text{COOH}$ and $^{13}\text{CH}_3^{13}\text{COOH}$) to demonstrate the isotope-tracer via ^1H NMR method. As shown in **Figure R6**, the split phenomenon of ^{13}C -linked hydrogen is effective for most single samples to distinguish the isotope. But we can also see that if there is no ^{13}C -linked hydrogen in liquid sample, such as the CH_3COOH and $\text{CH}_3^{13}\text{COOH}$, it is difficult to distinguish the ^{13}C -labeled samples.

Figure R6. The ^1H -NMR spectra of CH_3OH and $^{13}\text{CH}_3\text{OH}$ (a); $\text{CH}_3\text{CH}_2\text{OH}$, $\text{CH}_3^{13}\text{CH}_2\text{OH}$ and $^{13}\text{CH}_3^{13}\text{CH}_2\text{OH}$ (b); HCOOH and H^{13}COOH (c); and CH_3COOH , $\text{CH}_3^{13}\text{COOH}$ and $^{13}\text{CH}_3^{13}\text{COOH}$ (d).

Furthermore, we also carried out the ^1H NMR over a mixture of methanol, ethanol, formic acid and acetic acid. As shown in Figure R7, ^{13}C induced peak shifts and coupling in $^{13}\text{CH}_3^{13}\text{CH}_2\text{OH}$ and $^{13}\text{CH}_3^{13}\text{COOH}$ will also interfere each other due to the similar structure of the $^{13}\text{CH}_3$ -. Moreover, some of the most commonly used organic solvents (CH_3CN , DMF), sacrificial reagents (TOEA) and their derivatives will also increase the difficulty of detection when the authentic samples were measured by ^1H NMR strategy.

Figure R7. The ^1H -NMR spectra of a mixture of CH_3OH , $^{13}\text{CH}_3\text{OH}$, $\text{CH}_3\text{CH}_2\text{OH}$, $\text{CH}_3^{13}\text{CH}_2\text{OH}$, $^{13}\text{CH}_3^{13}\text{CH}_2\text{OH}$, HCOOH , H^{13}COOH , CH_3COOH , $\text{CH}_3^{13}\text{COOH}$ and $^{13}\text{CH}_3^{13}\text{COOH}$.

Overall, we are in favor of the reviewer's recommendation that we could mention the approach of ^1H -NMR is efficient for liquid product analysis through peak shifts and coupling constants, and the complementarity between different techniques could be highlighted. We revised the following sentences in our manuscript.

Over the years, various techniques using NMR spectroscopy and Fourier transform infrared spectroscopy (FT-IR) have been employed in isotope-tracer studies. Such as, the approach of ^1H -NMR is efficient for liquid product analysis through peak shifts and coupling constants of the ^{13}C -linked hydrogen; FT-IR equipped with a gas cell is effective for gaseous product analysis via the increased path length of a beam by multiple internal reflections.^{39,40}

We also added the following related reference in the citation.

39 Chatterjee, T., Boutin E. & Robert, M. Manifesto for the routine use of NMR for the liquid product analysis of aqueous CO_2 reduction: from comprehensive chemical shift data to formaldehyde quantification in water. *Dalton. Transactions* **49**, 4257-4265 (2020).

40 Kuhl, K. P. et al. Electrocatalytic Conversion of Carbon Dioxide to Methane and Methanol on Transition Metal Surfaces. *Journal of the American Chemical Society* **136**, 14107-14113 (2014).

Reviewer #4 (Remarks to the Author):

General comment

The author has made a sincere effort in this work to address the vital topic of the CO₂ photoreduction reaction. The author examined the scientific and challenging feature of the isotopic test in CO₂ reduction that has to be clarified to avoid misunderstanding. Consequently, they have effectively designed an experimental arrangement to deliver significantly greater reliability than the current experimental approach. However, much additional analysis and revisions are necessary. The fundamental problem is that the existing explanation in the paper lacks impact or importance concerning the topic. As a result, the authors must carefully handle this issue, and additional comments are provided below.

Comments

Response: Thank you very much for your kind and valuable comments. Your suggestions are helpful in improving the quality of the manuscript.

1. Several research in the literature demonstrate the applicability of SVUV-PIMS for CO₂ reduction; we recommend that the author discuss these studies in the introduction. A representative research paper is provided below.

Shao, Weiwei, et al. "In-plane heterostructure Ag₂S-In₂S₃ atomic layers enable boosted CO₂ photoreduction into CH₄." *Nano Research* 14.12 (2021): 4520-4527.

Response: Thank you very much for the suggestion. We have revised the introduction part as follow: "Even for the emerging synchrotron vacuum ultraviolet photoionization mass spectrometry (SVUV-PIMS) strategy, the existence of interfering factors induced by the long-time sampling is also uncertainty.⁵³ So the gas chromatograph (GC) is essential to be added to the sampling system to separate molecular species before collecting mass spectra (MS).^{54, 55}"

We have added these related references in our manuscript.

53 Shao, W. et al. In-plane heterostructure Ag₂S-In₂S₃ atomic layers enable boosted CO₂ photoreduction into CH₄. *Nano Research* 14, 4520-4527 (2021).

2. The author discussed the combination of ¹³CO₂, ¹³CO, and water vapour. Is there a particular rationale for selecting ¹³CO > 5% v/v? Additionally, if we alter the combination, v/v will affect the results; whether yes or no, please provide the pertinent information.

Response: Thanks for the comments. We choosing ¹³CO > 5% v/v is due to the currently highest yield of CO in usual CO₂ photoreduction process is much less than 5% v/v. So, the different ratio cannot be resolved in MS even ¹³CO > 5% v/v, the lower content of CO in the actual testing is impossible to be distinguished. When we further increased the content of CO (~ 30% v/v), as shown in **Figure R8**, the different ratio of m/z=29 is visible to the naked eye.

Figure R8 Comparison of MS spectra between $^{13}\text{CO}_2$ and $\sim 5\%$ v/v ^{13}CO in $^{13}\text{CO}_2$ (a) and $^{13}\text{CO}_2$ and $\sim 30\%$ v/v ^{13}CO in $^{13}\text{CO}_2$

3. The author claims that the formation of HO^+ and $^{13}\text{CO}^+$ raises questions on the study; is it possible to provide additional proof for this claim? In addition, HO^+ must, if possible, demonstrate the H_2O isotopic test.

Response: Thank you very much for your comment. As mentioned in the manuscript, both $^{13}\text{CO}_2$ and ^{13}CO can generate the peak of $m/z=29$ during the MS detection (Figure R9). For $^{13}\text{CO}_2$, the source of $^{13}\text{CO}^+$ is the unavoidable dissociation of $^{13}\text{CO}_2$; while the source of $^{13}\text{CO}^+$ in ^{13}CO is come from the ionization of ^{13}CO molecule.

Figure R9 The MS spectra of $^{13}\text{CO}_2$ (a) and ^{13}CO (b).

Moreover, we also carried out the MS detection of the H_2O , D_2O (99.9 atom % D) and H_2^{18}O (97 atom % ^{18}O). The experimental description of testing is as follow:

The samples were placed in headspace sampler, then the samples analyzed by gas chromatography-mass spectrometry (8890-5977B GC-MS instrument, Agilent Technologies,

USA) equipped with commercial capillary columns. The column was maintained at 120 °C for 15 min, and the flow of the carrier was 0.8 ml L⁻¹. The temperatures of the injector, EI source, and the GCITF were set to be 200, 200, and 250 °C, respectively. The mass-to-charge ratio of mass scanning mode were set from 2 to 70.

As show in **Figure R10**, all the water samples (H₂O, D₂O and H₂¹⁸O) exhibit similar peak at 2.35 min in the TIC spectra. Further comparison with the corresponding MS, it can be found that the H₂O exhibits three main peaks, those are m/z=18 (molecular ions of H₂O⁺), m/z=17 (fragments ions of HO⁺) and m/z=16 (fragments ions of O⁺); H₂¹⁸O also exhibits three main peaks, those are m/z=20 (molecular ions of H₂¹⁸O⁺), m/z=19 (fragments ions of H¹⁸O⁺) and m/z=16 (fragments ions of ¹⁸O⁺); Although the D₂O exhibits two more peaks of m/z=20 (molecular ions of D₂O⁺), m/z=19 (fragments ions of DO⁺), the three main peaks is almost the same to the H₂O. This is because the heat of evaporation and the boiling point of D₂O (41.60 kJ mol⁻¹, 101.42 °C) is a little bit higher than H₂O (40.67 kJ mol⁻¹, 100 °C) leading to the H₂O is much easily to be collected by headspace sampler.

Figure R10 The TIC spectra of the water sample collected from headspace sample (a) and the corresponding MS spectra recorded at 2.35 min in TIC (b).

Based on the analysis, the presence of dissociated ¹³CO₂ (generated fragments of ¹³CO⁺, m/z = 29) and vapor (generated fragments of HO⁺, m/z = 17) throw the current traceability of ¹³CO and ¹³CH₄ into doubt.

4. Several research on SVUV-PIMS have previously been published; thus, it is understandable that the author wants this analysis to investigate this technique. Nonetheless, it appears to be a typical analysis. We believe the significance of this analysis merits further investigation. This approach and existing modifications are missing connections. Thus, we suggest that the author must have coherence.

Response: Thank you very much for your comment. As an emerging strategy for the isotope-

tracer study, SVUV-PIMS merits further investigation. We have shown some results based on this strategy for the statement that a gas chromatograph (GC) is essential to be added to the sampling system to separate molecular species before collecting mass spectra (MS). Our logic is that although several research on SVUV-PIMS has previously been published, the existence of interfering factors induced by the long-time and large-amount sampling is still uncertain. We believe the brief discussion about SVUV-PIMS will underscore the importance of the approach that we have established.

5. There is considerable concern that HP-PLOT Molesieve Columns are commercially accessible; we appreciate the concept of putting HP-PLOT and Molesieve in parallel. Nevertheless, the author can elaborate on how this methodology will contribute to scientific importance.

Response: Thank you very much for your comment. Currently, there is no standard strategy to separate and analyze all these products, reactants, and air components simultaneously. Even for the most reported photocatalytic CO₂ conversion process of CO₂ to CO, a mixture of O₂, N₂, CO, CH₄, CO₂ and water vapor cannot be completely separated with one column due to the irreversible adsorption of CO₂ in the HP-Molesieve column and poor separation of CO/N₂ in the HP-PLOT/Q column. To solve this problem, we designed a parallel connection system containing both HP-Molesieve and HP-PLOT/Q columns.

6. Figure S21a photocatalytic system still shows the ¹⁶CH₄, is this surface contamination of photocatalyst? In Fe(III) porphyrin complex, while synthesising, all the precursors are organic, and also, there is no high-temperature treatment; how does the author avoid carbon contamination?

Response: Thank you very much for your comment. We showed the MS spectra of CH₄ from the photocatalytic system with Fe^{III} porphyrin complex as photocatalyst compared to the standard samples in Figure S21a. As shown, although there is a peak at m/z =16 in the MS spectra, it can be found that relative ratio (each substance has a specific ratio among the molecular ion peak and fragment ion peaks) among the molecular ion peaks (¹³CH₄⁺, m/z=17) and fragment ion peaks (¹³CH₃⁺, m/z=16; ¹³CH₂⁺, m/z=15; ¹³CH⁺, m/z=14; ¹³C⁺, m/z=13;) of the generated CH₄ present coherence to the isotope labeled standard ¹³CH₄ (Figure 4a, main manuscript) and present a mass shift effect of (M +1) versus the non-isotope labeled standard ¹²CH₄ (Figure 21a, Supplementary information). So, the m/z=16 in Figure S21a photocatalytic system is clearly attributed to the fragment ion peak of ¹³CH₃⁺, instead of the molecular ion peak of ¹²CH₄⁺. In other words, we can avoid carbon contamination of the photocatalyst and judge the CH₄ is really come from the CO₂ by this comparison.

7. In the case of the photocatalytic activity, it is time-dependent; for instance, is it possible to show the time-dependent GC-MS? In Figure S22, with light and without light radiation, how do we understand that the concentration of ¹³CO₂ changes and converts to ¹³CO?

Response: Thank you very much for your valuable suggestion. As mentioned by the

reviewers, the photocatalytic activity is time dependent. it is meaningful to develop a strategy on GC-MS for the quantitative research. but it is also very difficult that has never been reported. In fact, our group is conducting research on this topic. Although it is not perfect now, we still carried out the experiment based on the suggestion of the reviewer.

Firstly, we developed an online system and the standard curves for the time-dependent GC-MS. As shown in **Figure R11**, both the ^{13}CO and the $^{13}\text{CH}_4$ exhibit good linear relationship based on the GC-MS detection.

Figure R11 The TIC spectra (a) and the corresponding MS spectra (b) of standard ^{13}CO and $^{13}\text{CH}_4$ mixture sample; the established standard curves of ^{13}CO (c) and $^{13}\text{CH}_4$ (d) by GC-MS.

Subsequently, we carried out the time-dependent study of CO_2 photoreduction with GC-MS analysis. As shown in **Figure R12**, the generation of the ^{13}CO is increased with the time range from 2 to 10 hours (the photogenerated electron were used to consume residual O_2 in the system during the first 2 hours).

Figure R12 The time-dependent TIC spectra by using $Ru(bpy)_3Cl_2$ as photocatalyst (a) and the calculated amount of generated ^{13}CO by the established standard curve.

8. In the gas phase, we strongly suggest the added some result and discussion about the time-dependent study of CO_2 photoreduction with GC-MS analysis.

Response: Thank you very much for your valuable suggestion. As mentioned in the above response, we have developed an effective system for quantitative research, given that we have not found a reliable gas-solid phase system to carry out the time-dependent isotope experiments after many experiments. We hope to present the results of our lasted research about the quantitative isotope in future manuscripts.

9. In the case of liquid-phase CO_2 photoreduction, several photocatalyst systems employ organic solvents and sacrificial reagents containing organic moieties (Such as TEOS). Does the author believe that this will impede the separation of the product? Please include some experiments and an explanation, if feasible.

Response: Thank you very much for your comment. The sacrificial reagent of triethanolamine (TEOA) has a very high boiling point, it is difficult to volatilize to enter GC-MS. The organic solvent of acetonitrile (CH_3CN) is volatile easily and will enter the GC together with the gas sample. Based on our research, it takes a long time to go through the HP-PLOT/Q and HP-FFAP columns and then enter the mass spectrometer. We usually emptying the involved CH_3CN by a high temperature postrun program of GC-MS. So, it will not impede the separation of the products.

10. In many figures, captions are not correctly mentioned. Hence it is difficult to refer to the corresponding figure; please check the supporting information in the caption; there is no a), b) and so on mentioned.

Response: Thank you very much for your comment. We have carefully checked and revised

the figures and captions in main manuscript and supplementary information.

11. I recommend for the authors to study recently published papers related to this work.

- "Solar fuels: Research and development strategies to accelerate photocatalytic CO₂ conversion into hydrocarbon fuels", *Energy & Environmental Science* 15 (2022) 880 – 937

- "Electronic interaction between transition metal single-atoms and anatase TiO₂ boosts CO₂ photoreduction with H₂O", *Energy & Environmental Science* 15 (2022) 601-609

Response: Thank you very much for your comment. We have added these related references in our manuscript.

8 Gong, E. et al. Solar fuels: Research and development strategies to accelerate photocatalytic CO₂ conversion into hydrocarbon fuels. *Energy & Environmental Science* 15, 880-937 (2022).

18 Lee, B. H. et al. Electronic interaction between transition metal single-atoms and anatase TiO₂ boosts CO₂ photoreduction with H₂O. *Energy & Environmental Science* 15, 601-609 (2022).

REVIEWER COMMENTS

Reviewer #1 (Remarks to the Author):

I am pleased to see that the authors have carefully address all questions, added more isotopic data, and performed long activity tests. These additional results make this work more credible and complete. I recommend this work be published in Nature Communications upon addressing following questions.

1. The authors mentioned in their response that post-run allows CO₂ to be desorbed from the HP-Molesieve column. But the molecular sieve is alkaline, and CO₂ is acidic, which makes the adsorption of CO₂ almost permanent. Can the authors provide the method of post-run (temperature and time, etc.)? What is the frequency of operation of the post-run?
2. The authors clarify the sealing of the system, but I can't find the experimental data mentioned by the authors, can the authors indicate it? Also, could the authors provide the method of syringe injection (syringe model and volume of injected sample)?
3. In Figure R6d, I'm curious if the authors have marked the sample names incorrectly.
4. The authors used sacrificial agents in the later replicate experiments, which are also used in many photocatalysis works. However, the oxidation of the sacrificial agent may produce C₁ products or H₂ (DOI: 10.1021/acseenergylett.8b00196). the GC-MS as well as isotopic methods proposed by the authors can be a good way to avoid false positive results. Therefore, I strongly suggest that the authors emphasize the need for isotope experiments and GC-MS tests, especially for reaction systems in which sacrificial agents are used.

Reviewer #2 (Remarks to the Author):

In the revised version, the authors have fully addressed all my comments and concerns. The article can now be accepted for publication without additional changes.

Reviewer #3 (Remarks to the Author):

The authors have provided detailed and pertinent answers to the many queries of the reviewers, including mine, and I am satisfied with both the detail and accuracy level of these answers. In my view, the paper could now be published in Nat. Commun.

Reviewer #4 (Remarks to the Author):

In my opinion, the authors revised the manuscript well according to the reviewers' comments. However some part of English are not well written. It would be better to have English editing service.

Point-by-point response to the reviewers' comments.

Reviewer #1 (Remarks to the Author):

I am pleased to see that the authors have carefully address all questions, added more isotopic data, and performed long activity tests. These additional results make this work more credible and complete. I recommend this work be published in Nature Communications upon addressing following questions.

Response: Thank you very much for your positive comments and efforts in reviewing. Your suggestions are helpful in improving the quality of the manuscript.

1. The authors mentioned in their response that post-run allows CO₂ to be desorbed from the HP-Molesieve column. But the molecular sieve is alkaline, and CO₂ is acidic, which makes the adsorption of CO₂ almost permanent. Can the authors provide the method of post-run (temperature and time, etc.)? What is the frequency of operation of the post-run?

Response: Thank you very much for the comment. Although the adsorption of CO₂ is almost permanent, there are still some CO₂ could come out during post-run. We carried out the experiment to verify whether post-run make some CO₂ flow out from HP-Molesieve column. As shown in Figure R1, an irregular peak appeared in the TIC, when we raised the column temperature to 300 °C. This corresponding MS spectra showed that it could be attributed to the ¹³CO₂ and H₂O. We have added the details of the method of post-run in the Supporting Information and we operated the post-run after each injection.

Figure R1. The TIC and MS (inset) of CO₂ flowing from the HP-Molesieve column that recorded by GC-MS.

2. The authors clarify the sealing of the system, but I can't find the experimental data mentioned by the authors, can the authors indicate it? Also, could the authors provide the method of syringe injection (syringe model and volume of injected sample)?

Response: Thank you very much for the comment. We added the following experimental data that would prove the sealing of our system (Figure R2). The syringe model is the VICI Pressure-Lok Precision Analytical Syringe A-2 Series (050033), the volume of the injected sample is 0.5 mL each injection. We also provided the method of syringe injection in the Supporting Information.

Figure R2. The Air tightness detection of the sealed system during whole day.

3. In Figure R6d, I'm curious if the authors have marked the sample names incorrectly.

Response: Thank you very much for the comment. We have revised Figure R6d.

Figure R6. The ¹H-NMR spectra of CH₃OH and ¹³CH₃OH (a); CH₃CH₂OH, CH₃¹³CH₂OH and ¹³CH₃¹³CH₂OH (b); HCOOH and H¹³COOH (c); and CH₃COOH, CH₃¹³COOH and ¹³CH₃¹³COOH (d).

4. The authors used sacrificial agents in the later replicate experiments, which are also used in many photocatalysis works. However, the oxidation of the sacrificial agent may produce C1 products or H₂ (DOI: 10.1021/acsenergylett.8b00196). the GC-MS as well as isotopic methods proposed by the authors can be a good way to avoid false positive results. Therefore, I strongly suggest that the authors emphasize the need for isotope experiments and GC-MS tests, especially for reaction systems in which sacrificial agents are used.

Response: Thank you very much for the comment. We also note this important article from the editor-in-chief of the ACS Energy Lett. We strongly agree with the reviewer about the importance of isotope experiments for reaction systems in which sacrificial agents are used. In fact, isotope experiment has been carried out in the most of the reported reaction systems involved sacrificial agents. However, current discussion on isotope tracing experiments in these reported works are not rigorous and demonstrate where pitfalls and misunderstandings make isotope product traceability difficult. That is why we carried out the current work. Based on our research, we found that not only the systems containing sacrificial agents requires isotope traceability, but systems without sacrificial agents (Last round response Figure R5) also require this standard guidelines for isotope tracing experiments. All in all, we really appreciate the high recognition of our work by reviewer, we have emphasized the need for isotope experiments and accurate GC-MS tests is not only necessary for the reaction systems using sacrificial agents but also important for the reaction systems without sacrificial agents as follow:

“Moreover, it should be emphasized that the isotope experiments and accurate GC-MS tests is not only necessary for the reaction systems using sacrificial agents but also important for the reaction systems without sacrificial agents.”

Reviewer #2 (Remarks to the Author):

In the revised version, the authors have fully addressed all my comments and concerns. The article can now be accepted for publication without additional changes.

Response: Thank you very much for your positive comments and efforts in reviewing.

Reviewer #3 (Remarks to the Author):

The authors have provided detailed and pertinent answers to the many queries of the reviewers, including mine, and I am satisfied with both the detail and accuracy level of these answers.

In my view, the paper could now be published in Nat. Commun.

Response: Thank you very much for your positive comments and efforts in reviewing.

Reviewer #4 (Remarks to the Author):

In my opinion, the authors revised the manuscript well according to the reviewers' comments. However, some part of English are not well written. It would be better to have English editing service.

Response: Thank you very much for your positive comments and efforts in reviewing. We have revised the manuscript verbatim. All the changes in the manuscript text file were revised with track changes.

REVIEWERS' COMMENTS

Reviewer #1 (Remarks to the Author):

The authors have addressed my comments. The revised manuscript is recommended for publication.